# TEMPO: Temporal Multi-scale Autoregressive Generation of Protein Conformational Ensembles

**Yaoyao Xu**[§¶]**, Di Wang**[¶]**, Zihan Zhou**[§]**, Tianshu Yu**[§†]**, Mingchen Chen**[¶†]

[§] School of Data Science, The Chinese University of Hong Kong, Shenzhen
[¶] Changping Laboratory, Beijing
{yaoyaoxu,zihanzhou1}@link.cuhk.edu.cn, yutianshu@cuhk.edu.cn
{lotus,mingchenchen}@cpl.ac.cn

## Abstract

Understanding the dynamic behavior of proteins is critical to elucidating their functional mechanisms, yet generating realistic, temporally coherent trajectories of protein ensembles remains a significant challenge. In this work, we introduce a novel hierarchical autoregressive framework for modeling protein dynamics that leverages the intrinsic multi-scale organization of molecular motions. Unlike existing methods that focus on generating static conformational ensembles or treat dynamic sampling as an independent process, our approach characterizes protein dynamics as a Markovian process. The framework employs a two-scale architecture: a low-resolution model captures slow, collective motions driving major conformational transitions, while a high-resolution model generates detailed local fluctuations conditioned on these large-scale movements. This hierarchical design ensures that the causal dependencies inherent in protein dynamics are preserved, enabling the generation of temporally coherent and physically realistic trajectories. By bridging high-level biophysical principles with state-of-the-art generative modeling, our approach provides an efficient framework for simulating protein dynamics that balances computational efficiency with physical accuracy.

## 1 Introduction

The intersection of artificial intelligence and protein science has revolutionized our understanding of biological systems. Recent breakthroughs in AI for protein research have transformed structure and function prediction [32, 37, 46, 22], protein design [3, 13, 25], and interaction modeling [9, 14, 47, 30]. However, while static structural understanding has advanced dramatically, accurately modeling protein dynamics remains an outstanding challenge at the frontier of computational biology.

Protein dynamics are characterized by two fundamental properties. First, they are inherently hierarchical and multi-scale, with motions naturally separating into slow collective movements (nanoseconds to microseconds) that typically correspond to functionally relevant conformational changes, and fast local fluctuations (picoseconds to nanoseconds) that reflect atomic-level interactions [15, 17]. This multi-scale organization forms the theoretical foundation for analytical methods like Principal Component Analysis and Normal Mode Analysis [8], demonstrating that protein dynamics can be effectively decomposed into essential subspaces operating at different time scales. Second, protein motions exhibit strong temporal correlations, where the continuous evolution of conformational states follows specific pathways critical for biological functions. These temporally correlated dynamic ensembles have proven essential in understanding enzyme catalysis mechanisms [48], characterizing drug-binding pathways [7], and elucidating allosteric regulation [31]. For instance, recent studies

---

[†] Corresponding authors

39th Conference on Neural Information Processing Systems (NeurIPS 2025).

have demonstrated that analyzing dynamic ensembles can reveal cryptic binding sites that only become accessible through specific conformational transitions [5].

While molecular dynamics (MD) simulations can naturally capture both properties by solving Newton's equations of motion at atomic resolution, their computational demands make them impractical for large-scale applications. Even with specialized hardware and enhanced sampling techniques, MD simulations are typically limited to microsecond timescales and small protein systems, making it challenging to systematically study slow conformational changes or analyze large protein datasets [38]. Current generative approaches, particularly diffusion-based models [19, 23, 10], fundamentally fail to leverage these characteristics. These methods generate conformational ensembles by simultaneously producing and optimizing protein states independently, by learning an energy landscape rather than capturing the true sequential and multi-scale nature of protein dynamics. This approach cannot accurately represent the causal chain of events that governs protein conformational changes, limiting their ability to generate physically consistent trajectories.

Motivated by these challenges, we propose TEMPO – a multi-scale autoregressive framework that models and generates protein dynamics across different temporal scales. Our approach combines a low-resolution model capturing essential conformational transitions with a high-resolution model generating detailed local fluctuations, directly translating biophysical principles into a computational framework for generating physically realistic trajectories. Extensive experiments demonstrate that our method achieves significant improvements over existing approaches.

Our work makes several key contributions:

- **Algorithm Design**. TEMPO introduces a novel multi-scale framework that captures both protein collective motions and local fluctuations, enabling efficient trajectory generation orders of magnitude faster than MD simulations.

- **Performance and Metrics**. Our method achieves state-of-the-art performance in both structural accuracy and computational efficiency in various metrics, outperforming existing methods in matching MD ground truth while requiring fewer computational resources.

- **Extensive Analysis**. We demonstrate TEMPO's ability to capture biologically meaningful protein motions through comprehensive case studies and analyses.

## 2 Related Work

**Protein Ensemble Generation.** Recent advances in deep learning have revolutionized the generation of protein conformational ensembles. Traditional approaches rely on MSA subsampling with AlphaFold2 [32], which provides limited control over conformational diversity. Modern deep learning methods have introduced more sophisticated techniques. AlphaFlow [19] fine-tunes single-state predictors under a flow matching framework to generate protein conformational ensembles. ESM-Flow [19] extends this approach by leveraging protein language models. BioEMU [23] employs a diffusion-based framework to generate thermodynamically accurate ensembles. ConfDiff [43] incorporates force-guided networks with diffusion models to enhance generation fidelity, while Str2Str [27] introduces a structure-to-structure translation framework with roto-translation equivariance. However, these methods primarily focus on generating conformational ensembles that match equilibrium distributions, without explicitly modeling the temporal evolution of protein structures.

**Learning Molecular Dynamics.** Machine learning approaches have emerged as powerful tools for accelerating and enhancing molecular dynamics simulations. VAMPNet [28] pioneered the use of variational approaches for Markov processes in molecular kinetics. Recent works like DiffMD [45] employ diffusion models to estimate conformational density gradients, while DFF [4] establishes connections between score-based generative models and molecular force fields. The Distributional Graphformer (DiG) [50] predicts equilibrium distributions of molecular systems, enabling efficient conformational sampling. However, these methods often focus on general-purpose force field learning or small molecular systems, making them computationally intensive for large proteins.

**Multi-scale Dynamics Modeling.** The inherent multi-scale nature of protein dynamics has long been recognized in computational biology. Traditional MD analysis methods decompose protein motions into slow collective changes, which are crucial for biological function, and fast local fluctuations contribute to overall stability. Recent deep learning approaches have begun to address

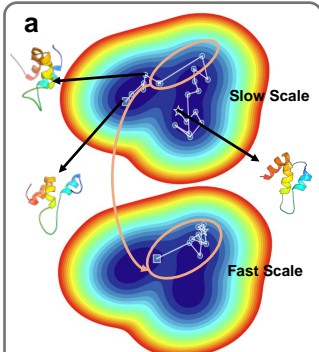 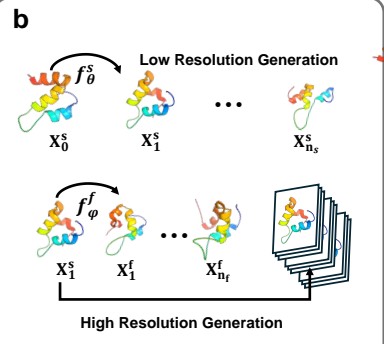 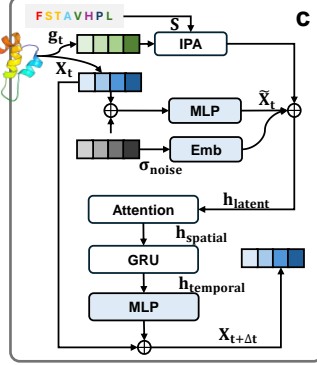

Figure 1: Overview of our multi-scale protein dynamics generation framework. **a)** The hierarchical free energy landscape of protein dynamics, where slow motions (upper) guide fast local fluctuations (lower). **b)** Our two-stage generation process: the low-resolution model $f_\theta^s$ captures slow collective motions, while the high-resolution model $f_\phi^f$ fills in detailed dynamics. **c)** The neural architecture that parameterizes both models features spatial-temporal encoding of protein conformations.

this multi-scale characteristic. EigenFold [20] models protein structures as systems of harmonic oscillators, naturally inducing a cascading-resolution generative process along system eigenmodes. FoldFlow [6] proposes a family of flow-based generative models on SE(3), where the continuous-time dynamics naturally capture multi-scale structural variations - from global conformational changes to local refinements - through different time scales of the flow evolution. ITO [36] learns transition density operators that allow conditioning on arbitrary timesteps, focusing on coarse-grained C$\alpha$ representations with exponential distribution sampling. However, most existing approaches focus on either ensemble generation or short-timescale dynamics, without explicitly bridging the gap between different temporal scales in protein motion.

**Autoregressive Models in Structural Biology.** While diffusion models have recently dominated protein structure generation, auto-regressive approaches are gaining traction for their ability to model temporally coherent and physically consistent dynamics. In the domain of bio-molecules, arDCA [41] applies a simple yet effective auto-regressive framework to model protein sequence distributions, capturing co-evolutionary couplings while enabling efficient sequence sampling and fitness prediction. For protein structure modeling, Structure Language Models [26] employ latent-space auto-regression to efficiently generate diverse backbone conformations, while equivariant models such as EquiJump [12] build an SO(3)-equivariant transport model that bridges long time intervals of all-atom protein MD by stochastically interpolating between snapshots. These advances demonstrate the promise of auto-regressive models in capturing complex bio-molecular dynamics across multiple timescales.

## 3 Method

### 3.1 Preliminaries

Given an initial protein structure $\mathbf{X}_0$ of sequence length $L$, our goal is to learn a generative model that produces protein backbone trajectories $\chi = [\mathbf{X_1}, \ldots \mathbf{X_T}]$, where each $\mathbf{X_i}$ represents the backbone conformation at time step $i$. While full-atom protein structure representation has been widely adopted [32, 23, 19], we focus on backbone dynamics as they capture the essential conformational changes that determine protein function. This choice is motivated by several key observations: **(1)** many important biological processes, such as protein folding and large-scale conformational changes, are primarily determined by backbone movements [33], **(2)** side-chain motions typically occur at faster timescales. They can be considered as local fluctuations around backbone configurations [16].

Consistent with standard representations in protein modeling [32], we describe each residue's backbone conformation using an SE(3) frame along with the torsion angles $(\phi, \psi, \omega)$:

$$\chi_t^l = ((R, \mathbf{t}), (\phi, \psi, \omega)), \quad \chi \in \left( \left[ \text{SE}(3) \times \mathbb{T}^3 \right]^L \right)^T \tag{1}$$

where $t$ denotes the time step and $l$ indicates the residue index. The SE(3) frame, representing the rigid body transformation, consists of a rotation component encoded as a unit quaternion from the positive real part $\hat{\mathbb{Q}}^+ \subset \mathbb{R}^4$ and a translation vector in $\mathbb{R}^3$, yielding a 7-dimensional representation. Each torsion angle is encoded as a 2D vector $[\sin(\theta), \cos(\theta)] \in S^2$ to avoid discontinuities at the periodic boundary. Combining these components, each residue at each time step is represented by a 13-dimensional vector:

$$\chi_t^j \in \left( \hat{\mathbb{Q}}^+ \oplus \mathbb{R}^3 \right) \times (S^2)^3 \subset \mathbb{R}^{13} \tag{2}$$

This representation captures the essential geometric features of protein backbone dynamics while maintaining computational efficiency, and the detailed data processing could be found in Appendix B.

## 3.2 SDE-based Protein Dynamics Modeling

Protein dynamics in solution naturally follows Langevin dynamics, which describes the motion of particles under both conservative forces and random collisions with solvent molecules [49]. The Langevin equation captures two key aspects of protein motion: **(1)** Deterministic forces arising from inter-atomic interactions that drive conformational changes, **(2)** Random forces from thermal fluctuations that contribute to the stochastic nature of protein dynamics [11].

Motivated by this physical principle, we model protein dynamics as a stochastic differential equation (SDE) process, which can be viewed as a continuous-time generalization of the Langevin dynamics. In our framework, the evolution of protein conformations follows a combination of deterministic drift and stochastic diffusion. Specifically, for a protein conformation $X_t$ at time $t$, its temporal evolution can be described as:

$$dX_t = \mu(X_t)dt + \sigma dW_t \tag{3}$$

where $\mu(X_t)$ represents the drift term that captures the deterministic dynamics, $\sigma$ is the diffusion coefficient, and $W_t$ denotes a standard Brownian motion. The drift term $\mu(X_t)$ is learned by our model, while the stochastic component is simulated through Gaussian noise injection.

In discrete time steps, our model approximates this continuous SDE process as:

$$X_{t+\Delta t} = X_t + f_\theta(X_t)\Delta t + \epsilon_t \sqrt{\Delta t} \tag{4}$$

where $f_\theta$ is our neural network model parameterized by $\theta$ that learns the drift dynamics, $\Delta t$ is the time step, and $\epsilon_t \sim \mathcal{N}(0, \sigma^2 I)$ represents the Gaussian noise. This formulation allows our model to capture both the deterministic conformational changes and the stochastic nature of protein dynamics. This approach is theoretically justified as we learn the conditional expectation $\mathbb{E}[X_{t+1}|X_t]$ in the finite time-step regime, mirroring how numerical MD integrators operate with deterministic updates plus controlled stochastic components for temperature regulation [49, 11].

However, protein dynamics typically exhibits non-Markovian behavior at short time scales [18], meaning that future states depend on multiple previous states rather than just the current one. To account for this memory effect, we extend our model to consider multiple timesteps:

$$\mathbf{X}_{t+1:t+2} = f_\theta(\mathbf{X}_{t-1:t})\Delta t + \epsilon_t \sqrt{\Delta t} \tag{5}$$

where $\mathbf{X}_{t-1:t}$ represents two consecutive frames at times $t-1$ and $t$, and the model predicts the next two frames $\mathbf{X}_{t+1:t+2}$. This design choice is supported by prior research in physical system modeling [51] and latent ODEs [34], where incorporating temporal memory has been shown to significantly improve the accuracy of dynamical predictions.

## 3.3 Multi-scale Dynamics Learning

Building upon the SDE-based framework, we decompose protein motion into a bio-physically motivated two-timescale formulation to capture both slow collective motions and fast local fluctuations[15, 17]. This decomposition is realized through coupled stochastic differential equations:

$$d\mathbf{X}_t^s = \mu_s(\mathbf{X}_t^s)dt + \sigma_s dW_t^s \tag{6}$$

$$d\mathbf{X}_t^f = \mu_f(\mathbf{X}_t^f)dt + \sigma_f dW_t^f \tag{7}$$

where $\mathbf{X}_t^s$ and $\mathbf{X}_t^f$ represent protein conformations at slow and fast timescales, respectively, with corresponding drift terms $\mu_s$ and $\mu_f$, and independent Brownian motions $W_t^s$, $W_t^f$.

To learn these multi-scale dynamics, we employ a unified neural architecture that operates at both timescales. Specifically, the drift terms $\mu_s$ and $\mu_f$ are parameterized by the same spatiotemporal encoder architecture $f$ but trained separately to capture scale-specific features. The discrete time evolution at each scale follows:

$$\mathbf{X}_{t+\Delta t_s}^s = \mathbf{X}_t^s + f_\theta^s(\mathbf{X}_t^s)\Delta t_s + \epsilon_t^s\sqrt{\Delta t_s} \tag{8}$$

$$\mathbf{X}_{t+\Delta t_f}^f = \mathbf{X}_t^f + f_\phi^f(\mathbf{X}_t^f)\Delta t_f + \epsilon_t^f\sqrt{\Delta t_f} \tag{9}$$

where $\Delta t_s$ and $\Delta t_f$ represent the time steps for slow and fast dynamics, with neural networks $f_\theta^s$ and $f_\phi^f$ learning the respective drift dynamics.

During inference, we employ a hierarchical sampling strategy that mirrors the natural organization of protein dynamics. Starting from an initial conformation $\mathbf{X}_0^s$, we first generate a sparse trajectory $\{\mathbf{X}_0^s, \mathbf{X}_{\Delta t_s}^s, ..., \mathbf{X}_{n_s\Delta t_s}^s\}$ using the slow-scale model $f_\theta^s$, which captures collective motions like domain reorientations. These slow-scale conformations then serve as anchoring states for the fast-scale model $f_\phi^f$, which generates the complete fine-grained trajectory $\{\mathbf{X}_t^f : t \in [0, n_f\Delta t_f]\}$, ensuring that fast local dynamics remain consistent with the broader conformational changes.

## 3.4 Spatiotemporal Protein Encoder

Here we detail the neural architecture that parameterizes the drift dynamics at both timescales. Our encoder design captures both spatial relationships between protein residues and temporal correlations in conformational dynamics. As illustrated in Figure 1, the network comprises three functional components: input representation, spatial-temporal encoding, and conformational prediction.

The input representation module processes protein conformations $\mathbf{X}_t \in \mathbb{R}^{L\times 13}$ with added noise terms that model the stochastic nature of protein dynamics. Specifically, we sample a noise scale $\sigma_{\text{noise}} \sim \mathcal{U}(a, b)$ and generate Gaussian noise $\epsilon \sim \mathcal{N}(0, \mathbf{I})$ to obtain noisy conformations $\tilde{\mathbf{X}}_t = \mathbf{X}_t + \sigma_{\text{noise}}\epsilon$. In parallel, protein sequences are embedded into feature vectors $\mathbf{s} \in \mathbb{R}^{L\times d_{\text{hidden}}}$ through a learnable embedding layer. These sequence features are combined with frame representations $\mathbf{g}_t \in \text{SE}(3)^L$ and noise scale in an Invariant Point Attention (IPA) [32] module to capture geometric relationships. The final latent representation is computed as:

$$\mathbf{h}_{\text{latent}} = \text{MLP}(\tilde{\mathbf{X}}_t) + \text{IPA}(\mathbf{g}_t, \mathbf{s}, \sigma_{\text{noise}}) + \text{Embed}(\sigma_{\text{noise}}) \tag{10}$$

where $\text{MLP} : \mathbb{R}^{13} \to \mathbb{R}^{d_{\text{hidden}}}$ projects the input features to hidden dimension $d_{\text{hidden}}$, and $\text{Embed}$ maps the scalar noise intensity to a $d_{\text{hidden}}$-dimensional vector. The latent representation then undergoes spatial-temporal processing through:

$$\mathbf{h}_{\text{spatial}} = \text{MultiHeadAttention}(\mathbf{h}_{\text{latent}}) \tag{11}$$

$$\mathbf{h}_{\text{temporal}} = \text{GRU}(\mathbf{h}_{\text{spatial}}) \tag{12}$$

The spatial module captures inter-residue interactions while the temporal module encodes frame-to-frame dependencies. The output module generates conformational updates:

$$\mathbf{X}_{t+\Delta t} = \mathbf{X}_t + \text{MLP}(\mathbf{h}_{\text{temporal}}) \tag{13}$$

where $\text{MLP} : \mathbb{R}^{d_{\text{hidden}}} \to \mathbb{R}^{13}$ maps the latent features back to the conformational space.

The training objectives for both timescales follow the same formulation, consisting of two terms: a reconstruction loss measuring the mean squared error between predicted and ground truth conformations and a physical constraint loss penalizing steric clashes between backbone atoms:

$$\mathcal{L}_{\text{total}} = \|\mathbf{X}_{t+\Delta t} - \hat{\mathbf{X}}_{t+\Delta t}\|^2 + \lambda \sum_{i\neq j} \text{ReLU}(1.2\text{Å} - \|r_i - r_j\|) \tag{14}$$

where $r_i$ and $r_j$ are positions of backbone atoms from different residues, and the minimum distance threshold of 1.2Å is chosen following the widely adopted steric criteria in Rosetta [1]. This objective is applied independently to train the slow and fast dynamics models, with appropriate time intervals $\Delta t_s$ and $\Delta t_f$, respectively.

# 4 Experiments

## 4.1 Experimental Settings

**Datasets.** We conduct experiments on two comprehensive molecular dynamics datasets: md-CATH [29] and ATLAS [42]. For mdCATH, we randomly sampled 1,000 proteins and their 320K temperature trajectories with three independent seeds for training. Each protein sequence was truncated to 240 residues, and trajectories were standardized to 400 frames at 1ns intervals through periodic extension or truncation. We randomly selected 50 proteins for validation and 64 proteins for testing, ensuring no overlap with the training set. For ATLAS, we follow the data split and processing protocol established by MDGen [21]. We rigorously quantified sequence similarity using mmseqs2, finding an average of 18.93% sequence similarity between training and test sets for mdCATH and 18.3% for ATLAS, well below standard thresholds (40%) for sequence relatedness.

**Baselines.** To evaluate our method's capability in capturing both protein dynamics and ensemble properties, we conduct comprehensive comparisons with four state-of-the-art baselines: BioEMU [23], AlphaFlow and ESMFlow [19], and MDGen [21]. While the first three methods primarily focus on generating conformational ensembles, MDGen, though not explicitly modeling dynamics, captures temporal evolution through training on the ATLAS dataset [42].

**Implementation details.** Our multi-scale modeling approach captures protein dynamics at two temporal resolutions. The low-resolution model generates trajectories at 20ns intervals, characterizing major structural transitions, while the high-resolution model operates at 1ns resolution to capture local fluctuations. We empirically chose the 20ns/1ns hierarchy based on established biophysical principles where the 20ns interval effectively captures major conformational transitions between different states in the free energy surface as visualized in Figure 1a, while the 1ns resolution represents the dataset's finest temporal sampling. The generation process follows a hierarchical strategy. The low-resolution model first produces a sequence of conformational states $\{X_t^s\}_{t=1}^{20}$ at $\Delta t = 20$ns. Specifically, each high-resolution segment is initialized by the corresponding low-resolution state ($X_{t+\Delta t}^f = f_\theta(X_t^s)$), followed by autoregressive sampling ($X_{t+k\Delta t}^f = f_\theta(X_{t+(k-1)\Delta t}^f)$ for $k = 2, ..., 20$ where $\Delta t = 1$ns). This hierarchical process generates a complete trajectory while maintaining consistency across different temporal scales.

During training, both scale-models simulate the forward process of protein dynamics SDE through autoregressive sampling with noise scales uniformly sampled from $[0.01, 0.05]$. At inference time, while the low-resolution model maintains similar noise levels, we increase the noise scale to $5.0$ for the high-resolution model. This elevated noise level enables diverse conformational sampling on the learned energy surface.

**Evaluation Framework.** Our comprehensive evaluation framework encompasses both ensemble properties and trajectory-specific characteristics. Following AlphaFlow, we analyze conformational flexibility through several complementary measures: **Dynamic Range** (the average C$\alpha$-RMSD between pairs of conformations within each ensemble, quantifying the overall conformational space explored), **Local Flexibility** (assessed through root mean square fluctuation (RMSF) analysis of atomic positions), and **Distribution Accuracy** (quantified using the root mean Wasserstein distance (RMWD) between predicted and ground truth conformational distributions).

The trajectory accuracy is measured by the backbone RMSD error between generated and ground truth trajectories relative to the native structure: $\text{Error}_{\text{frame}} = |\text{RMSD}_{\text{pred}} - \text{RMSD}_{\text{gt}}|$. This metric, averaged across all frames and test proteins, quantifies the model's ability to capture conformational change magnitudes accurately [24]. Further biological validation includes contact dynamics analysis, where contacts between C$\alpha$ atoms (8Å threshold) are classified as weak (initially present but dissociate in $> 10\%$ of the ensemble) or transient (initially absent but form in $> 10\%$ of the ensemble), with accuracy evaluated using Jaccard similarity between predicted and ground truth contact sets. Computational efficiency is assessed through the average inference time per protein in generating 400 snapshots. Additionally, we calculate the clash ratio, defined as the proportion of conformations containing steric clashes among the 400 generated snapshots for each protein. Detailed definitions of all metrics are provided in Appendix D.

Table 1: **Evaluation on mdCATH**. Comparing predicted ensembles with MD ensembles across various metrics. For protein flexibility and RMSF, ground truth values are in parentheses. Median values across 64 test ensembles are reported. The rightmost column shows TEMPO's performance on the up-sampling task. $r$: Pearson correlation; $J$: Jaccard similarity; $\mathcal{W}_2$: 2-Wasserstein distance.

| Metrics | TEMPO | BioEMU | AlphaFLOW-MD | ESMFLOW-MD | MDGEN | TEMPO(Up) |
|---|---|---|---|---|---|---|
| Pairwise RMSD($= 3.26$) | **2.78** | 13.82 | 2.00 | 2.32 | 1.11 | 3.06 |
| Pairwise RMSD $r \uparrow$ | **0.77** | -0.02 | 0.41 | 0.26 | 0.71 | 0.99 |
| All-atom RMSF($= 1.64$) | **1.60** | 10.08 | 0.99 | 1.18 | 0.56 | 1.64 |
| Global RMSF $r$ | **0.67** | 0.13 | 0.41 | 0.34 | 0.67 | 0.99 |
| Root mean $\mathcal{W}_2 \downarrow$ | 4.21 | 10.70 | 5.62 | 4.08 | **3.36** | 1.06 |
| MD PCA $\mathcal{W}_2 \downarrow$ | **2.33** | 2.49 | 2.38 | 2.36 | 2.62 | 0.63 |
| % PC-sim $> 0.5 \uparrow$ | 7.81 | 9.38 | 21.88 | **25.00** | 17.19 | 95.31 |
| Weak contacts $J \uparrow$ | 0.43 | 0.38 | 0.42 | **0.51** | 0.41 | 0.83 |
| Trans. contacts $J \uparrow$ | 0.20 | 0.12 | 0.27 | **0.28** | 0.20 | 0.60 |
| % Clash ratio $\downarrow$ | 4.75 | 19.7 | 15.5 | 5.23 | **0.42** | 4.12 |
| RMSD Error $\downarrow$ | **1.78** | - | - | - | 3.76 | 0.60 |
| Inference time (hour) | **0.006** | 0.25 | 4.5 | 4.7 | 0.008 | - |

## 4.2 Results and Analysis

We evaluate our framework on both mdCATH and ATLAS datasets. Table 1 presents results on mdCATH, while Table 2 shows results on ATLAS where all methods are trained and tested following MDGen's protocol. TEMPO demonstrates consistent strong performance across both datasets, achieving state-of-the-art results on key metrics.

Table 2: **Evaluation on ATLAS.** All methods trained and tested following MDGen's protocol.

| Metrics | BioEMU | AlphaFlow-MD | MDGen | TEMPO |
|---|---|---|---|---|
| Pairwise RMSD $r \uparrow$ | -0.02 | 0.48 | 0.48 | **0.91** |
| Global RMSF $r \uparrow$ | 0.09 | 0.60 | 0.50 | **0.89** |
| Root mean W2 $\downarrow$ | 19.23 | 2.61 | 2.69 | **1.49** |
| MD PCA W2 $\downarrow$ | 3.61 | 1.52 | 1.89 | **0.60** |
| % PC-sim $> 0.5 \uparrow$ | 14 | 44 | 10 | **76** |
| Weak contacts $J \uparrow$ | 0.26 | 0.51 | 0.62 | **0.74** |
| Trans. contacts $J \uparrow$ | 0.06 | 0.29 | **0.41** | 0.38 |
| RMSD Error $\downarrow$ | - | - | 3.20 | **1.83** |

Our multi-scale framework demonstrates distinct advantages across three critical dimensions of protein dynamics modeling. On ATLAS, TEMPO achieves substantial improvements over baselines, with Pearson correlation of 0.91 for pairwise RMSD and 0.89 for global RMSF, significantly outperforming other methods. The model captures 76% of principal components with similarity greater than 0.5, compared to 44% for AlphaFlow and 10% for MDGen, demonstrating superior ability to preserve essential collective motions. On mdCATH, in structural flexibility metrics, TEMPO achieves the closest match to molecular dynamics (MD) ground truth, with pairwise $C\alpha$-RMSD and residue-level RMSF closely matching MD values. The strong Pearson correlation ($r = 0.77$) between predicted and ground truth RMSD values reveals our hierarchical SDE formulation preserves the intrinsic roughness of protein energy landscapes.

In distribution matching metrics, TEMPO achieves the lowest MD PCA Wasserstein distance (0.60 on ATLAS, 2.33 on mdCATH) across both datasets, suggesting accurate preservation of principal motion patterns. While MDGen shows slight advantages in average conformational Wasserstein distance on mdCATH, our approach achieves better backbone RMSD error on both datasets (1.83 on ATLAS, 1.78 on mdCATH compared to MDGen's 3.20 and 3.76). Furthermore, our physically constrained learning reduces steric clashes compared to ESMFlow, BioEMU, and AlphaFlow, validating the biological plausibility of generated conformations.

In computational efficiency, TEMPO generates complete 400-frame trajectories in approximately 22 seconds, significantly faster than AlphaFlow and ESMFlow. Among all baselines, only MDGen is specifically designed for trajectory generation, yet it requires prohibitive computational resources. In

contrast, our multi-scale decomposition enables training on a single NVIDIA A100 GPU, with both slow-scale and fast-scale models operating within memory constraints.

These results collectively demonstrate that TEMPO's physics-informed multi-scale design achieves superior performance in conformational accuracy, temporal coherence, and computational efficiency across different datasets. Our experimental validation incorporates four analyses to verify TEMPO's spatiotemporal modeling: (1) Collective motion analysis to evaluate the capture of functionally relevant slow motions, (2) Up-sampling evaluates high-resolution reconstruction of local fluctuations, (3) State transition tracking examines temporal pathway fidelity, and (4) Free energy surface analysis to compare conformational sampling strategies. We focus subsequent analysis and visualizations on mdCATH as its free energy landscapes contain more distinct energy basins with clearer conformational transition pathways, better suited for analyzing dynamics modeling. Visual examples of generated protein trajectories are provided in Appendix E, offering an intuitive demonstration of our model's capability to capture protein dynamics. Additional ablation studies and detailed analyses are provided in the Appendix. All analyzed cases are identified by their PDB IDs.

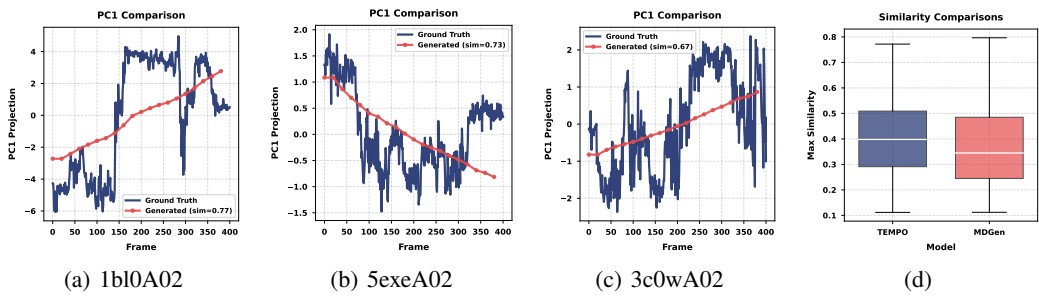

| (a) 1bl0A02 | (b) 5exeA02 | (c) 3c0wA02 | (d) |

Figure 2: Comparison of PC projections between MD and generated conformations. (a-c) PC1 trajectories from MD (blue) and our slow-scale trajectories (red) for three representative proteins. (d) Box plot of cosine similarity scores comparing TEMPO and MDGen across the test set.

**Slow Motion generation.** Protein slow motions, occurring on microsecond to millisecond timescales, often correspond to functionally relevant conformational changes such as domain movements [17]. Capturing these collective motions is challenging due to their long timescales and coordinated nature. We evaluate our model's capability in capturing such motions through principal component analysis (PCA), as the first few PCs typically describe the dominant collective motions in protein dynamics [2]. For representative cases (Figure 2(a-c)), we projected both MD trajectories and our generated slow-scale conformations onto the first principal component (PC1) of MD trajectories. The cosine similarity between these projections ranges from 0.67 to 0.77, indicating strong alignment of collective motions. Extending this analysis across all test proteins (Figure 2(d)), we evaluated the maximum cosine similarity between the first two PCs of generated ensembles and MD trajectories to assess the capture of collective motions. TEMPO achieves better performance than MDGen, with a mean similarity of 0.41 compared to MDGen's 0.36, suggesting potential for capturing slow motions while highlighting the challenging nature of this task.

**Up-sampling.** We evaluated our high-resolution model's capacity to capture detailed protein motions through up-sampling experiments, using ground truth low-resolution protein conformation as input. The conformational sampling quality was quantitatively assessed via free energy surface (FES) analysis in a reduced dimensionality space, obtained through PCA of protein backbone coordinates using Prody. The free energy landscapes were constructed by projecting conformations onto the first two principal components, followed by kernel density estimation and Boltzmann inversion at 300K. Figure 3 demonstrates the FES contour plots for two representative test proteins, comparing the distributions between MD ensembles and our generated trajectories. The close correspondence in FES characteristics, particularly in the location and depth of energy minima, validates our model's ability. More evaluation metrics across our test set (Table 1) further confirm that our high-resolution model effectively captures local protein fluctuations consistent with MD simulations. Additional case studies with extended protein sets are presented in Appendix I.

**State Transition.** After evaluating the two scale models separately, we further investigated TEMPO's ability to capture complete conformational transition pathways. Using four representative proteins

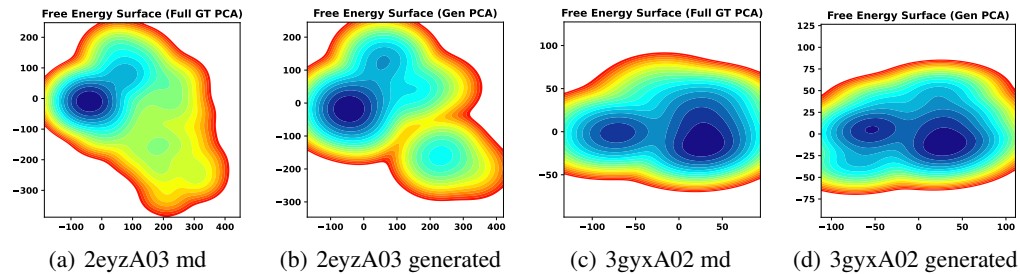

(a) 2eyzA03 md     (b) 2eyzA03 generated     (c) 3gyxA02 md     (d) 3gyxA02 generated

Figure 3: FES comparison between MD trajectories (a,c) and generated ensembles (b,d) for proteins 2eyzA03 (a,b) and 3gyxA02 (c,d). Colors represent free energy values from low (blue) to high (red).

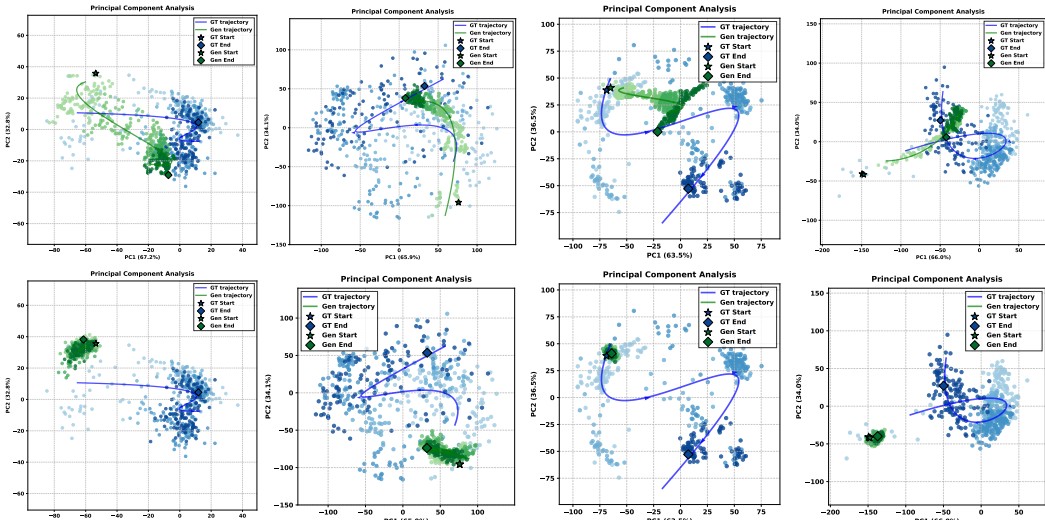

Figure 4: Comparison of conformational transitions in PC space between TEMPO and MDGen baseline (bottom). Ground truth MD trajectories are shown in blue, while generated trajectories are in green. The polynomial fitting curves highlight the temporal evolution of conformational changes (Protein from left to right: 2e9xB01, 1s79A00, 1bl0A02, 3cx5E01).

from our test set, we analyzed how well our integrated framework reproduces the sequential nature of conformational changes. We first constructed the PCA space using MD trajectories as reference, then projected both our generated trajectories and MD trajectories onto the first two principal components to visualize the conformational landscape. As shown in Figure 4, the upper row demonstrates TEMPO's ability to generate trajectories (green) that follow similar transition pathways as MD simulations (blue). In contrast, the MDGen tends to generate clustered conformations in limited regions of the PC space, failing to capture the full range of transitions. This comparison highlights the advantage of our temporal modeling approach over the frame-independent generation strategy, particularly in reproducing the sequential nature of conformational changes. The polynomial fitting curves further illustrate how our model better tracks the temporal evolution of these state transitions. Additional transition pathway analyses for an extended set of test proteins are provided in Appendix J.

**Free Energy Surface Coverage.** Our quantitative metrics effectively assess conformational stability and structural accuracy, but provide limited insight into the comprehensive exploration of conformational space. To address this, we established a reference framework using PCA derived from MD ensembles of four randomly selected proteins, subsequently constructing the corresponding free energy surface (FES). Projection of generated conformations onto this FES revealed that ESMFlow achieves broader coverage of the conformational landscape (Figure 5), consistent with its design for independent sampling across the energy surface.

This divergence reflects fundamentally different modeling objectives. ESMFlow optimizes conformational diversity through independent sampling, valuable for exploring thermodynamically accessible

states. In contrast, TEMPO's focused sampling is a design feature that preserves temporal correlations and physical constraints governing real protein motion. Real proteins must respect energy barriers and cannot instantaneously jump between distant conformations; biological processes depend on sequential, pathway-dependent conformational changes where kinetic accessibility differs from thermodynamic accessibility. Such trajectory-aware modeling ultimately serves our primary objective of elucidating the kinetic mechanisms underlying biological function, rather than maximizing configurational sampling (additional analyses in Appendix K).

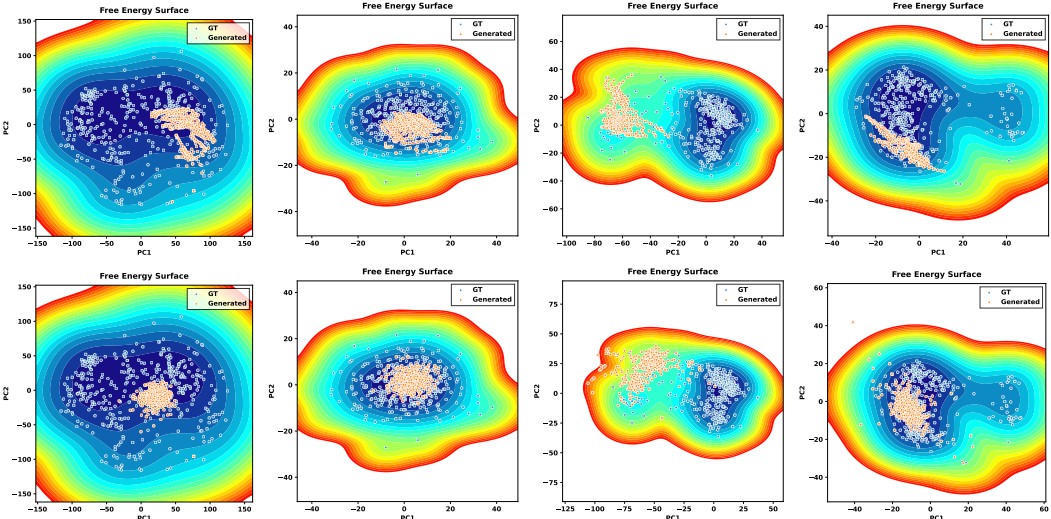

Figure 5: FES comparison between TEMPO (top row) and ESMFlow (bottom row) on four randomly selected test proteins. TEMPO's dynamic modeling shows more focused exploration of conformational space, whereas ESMFlow's independent sampling achieves broader coverage across the PCA-derived free energy surface (Protein from left to right: 1s79A00, 1x4tA01, 2e9xB01, 5b58T02).

## 5 Conclusion

In conclusion, our proposed TEMPO framework represents a significant advancement in the modeling and generation of protein dynamics by effectively addressing the inherent complexities of hierarchical and multi-scale behavior. Through the integration of a multi-scale autoregressive approach with stochastic differential equations, TEMPO successfully captures both slow collective motions and fast local fluctuations that characterize protein dynamics. Our comprehensive experimental validation demonstrates that TEMPO achieves superior performance across multiple metrics, from structural flexibility to distribution matching, while maintaining computational efficiency compared to existing methods. As we move forward, TEMPO's innovative design and proven capabilities hold the potential to facilitate further research in protein dynamics, ultimately contributing to a more comprehensive understanding of biological systems and their underlying mechanisms.

## 6 Acknowledgment

This work was supported in part by the National Science and Technology Major Project of China under Grant 2022ZD0116408. This work is also supported in part by the Guangdong Provincial Key Laboratory of Mathematical Foundations for Artificial Intelligence (2023B1212010001). We express our gratitude to Changping Laboratory for their research funding and support. The resources and collaborative environment provided have been instrumental in enabling us to achieve the objectives of this project.

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

## A  Limitation

While TEMPO demonstrates promising capabilities in protein dynamics generation, several aspects warrant further exploration. The current model shows limited generalization capability to unseen proteins, especially for cases involving large conformational changes. This could be addressed through expanded training data incorporating diverse protein architectures and enhanced model designs for capturing large-scale motions. Our two-scale temporal decomposition, though effective for the demonstrated trajectory lengths, might benefit from auto-regressive resolution mechanisms similar to those successful in image generation [40] when modeling longer timescale dynamics.

The framework's current limitation to single-protein backbone dynamics could be extended in several directions. Incorporating side-chain reconstruction and multi-molecular interactions could enable modeling of protein-ligand binding dynamics and DNA recognition processes, critical downstream applications in drug discovery and biological mechanism studies. Additionally, while existing metrics provide useful validation, developing more biologically-grounded evaluation protocols could better assess trajectory quality through direct correlation with experimental observables and functional outcomes. These directions collectively suggest rich potential for expanding both the scope and practical utility of deep learning-based protein dynamics modeling.

## B  Protein Structure Tokenization

We represent protein structures using a combination of local reference frames and torsion angles, resulting in a rotation and translation equivariant representation. For a protein with $L$ amino acids, our tokenization procedure is as follows:

**SE(3) Frame Representation.**  For each residue $i$, we construct a local reference frame using the backbone atoms (N, C$\alpha$, C) following the approach similar to AlphaFold [32]. Specifically:

- The origin $O_i$ is placed at the C$\alpha$ atom position
- The x-axis $\hat{x}_i$ is aligned with the normalized C$\alpha$-N bond vector
- The temporary vector $\vec{v}_i$ is the normalized C$\alpha$-C bond vector
- The z-axis $\hat{z}_i$ is computed as $\hat{z}_i = \frac{\hat{x}_i \times \vec{v}_i}{|\hat{x}_i \times \vec{v}_i|}$
- The y-axis $\hat{y}_i$ completes the right-handed coordinate system: $\hat{y}_i = \hat{z}_i \times \hat{x}_i$

Each SE(3) frame consists of a rotation matrix $R_i \in \mathrm{SO}(3)$ and a translation vector $t_i \in \mathbb{R}^3$:

$$R_i = [\hat{x}_i, \hat{y}_i, \hat{z}_i] \in \mathbb{R}^{3\times3}, \quad t_i = O_i \in \mathbb{R}^3 \tag{15}$$

To obtain a compact representation, we convert the rotation matrix to a unit quaternion $q_i \in \mathbb{S}^3$ following [39]:

$$q_i = \begin{bmatrix} q_{w,i} \\ q_{x,i} \\ q_{y,i} \\ q_{z,i} \end{bmatrix} = \begin{bmatrix} \frac{1}{2}\sqrt{1 + R_i[0,0] + R_i[1,1] + R_i[2,2]} \\ \frac{R_i[2,1] - R_i[1,2]}{4q_{w,i}} \\ \frac{R_i[0,2] - R_i[2,0]}{4q_{w,i}} \\ \frac{R_i[1,0] - R_i[0,1]}{4q_{w,i}} \end{bmatrix} \tag{16}$$

Combined with the translation vector, this yields a 7-dimensional vector $[q_{w,i}, q_{x,i}, q_{y,i}, q_{z,i}, t_{x,i}, t_{y,i}, t_{z,i}]$ for each residue, resulting in a tensor of shape $[L, 7]$ for the entire protein.

**Torsion Angle Representation.**  We complement the SE(3) frames with the backbone torsion angles $(\phi, \psi, \omega)$, which define the protein's conformation. For residue $i$, these angles are defined as:

$$\phi_i = \mathrm{dihedral}(C_{i-1}, N_i, C\alpha_i, C_i) \tag{17}$$
$$\psi_i = \mathrm{dihedral}(N_i, C\alpha_i, C_i, N_{i+1}) \tag{18}$$
$$\omega_i = \mathrm{dihedral}(C\alpha_i, C_i, N_{i+1}, C\alpha_{i+1}) \tag{19}$$

Rather than using the raw angles, we represent each angle as a 2D vector $[\sin(\theta), \cos(\theta)]$ to avoid discontinuities at $\pm\pi$, as commonly done in protein structure prediction models [44]. This results in a 6-dimensional vector per residue:

$$v_{\text{torsion},i} = [\sin(\phi_i), \cos(\phi_i), \sin(\psi_i), \cos(\psi_i), \sin(\omega_i), \cos(\omega_i)] \quad (20)$$

This yields a tensor of shape $[L, 6]$ for the protein's torsion information.

**Final Representation.** We concatenate the SE(3) frame and torsion angle representations to obtain a comprehensive protein structure encoding of shape $[L, 13]$, where a 13-dimensional vector represents each residue:

$$v_i = [q_{w,i}, q_{x,i}, q_{y,i}, q_{z,i}, t_{x,i}, t_{y,i}, t_{z,i}, \sin(\phi_i), \cos(\phi_i), \sin(\psi_i), \cos(\psi_i), \sin(\omega_i), \cos(\omega_i)] \quad (21)$$

**Rotation and Translation Equivariance.** Our final protein representation—comprising unit quaternions and translation vectors for SE(3) frames and backbone torsion angles is equivariant to global rigid-body motions. Specifically, a global rotation $R_g \in \text{SO}(3)$ and translation $t_g \in \mathbb{R}^3$ transform each local frame $(R_i, t_i)$ as $(R_g R_i, R_g t_i + t_g)$. When using unit quaternions $q_i$ to represent $R_i$, this corresponds to a left quaternion multiplication:

$$q_i' = q_g \otimes q_i, \quad t_i' = R_g t_i + t_g$$

where $q_g$ is the unit quaternion corresponding to $R_g$, and $\otimes$ denotes quaternion multiplication.

The torsion angles $(\phi, \psi, \omega)$, represented as $[\sin(\theta), \cos(\theta)]$ pairs, are internal degrees of freedom and remain invariant under global SE(3) transformations. Therefore, our 13-dimensional representation $v_i$ is globally equivariant and captures both spatial orientation and internal conformation. This formulation ensures that the learned model respects 3D geometric symmetries, consistent with the principles of equivariant neural networks [35].

# C Algorithm

Algorithm 1 describes our multi-scale training procedure, which is applied to both low-resolution and high-resolution models. The algorithm implementation differs in how we prepare the training trajectories for each scale:

**Low-resolution Training.** For the slow-scale model ($\Delta t_s = 20\text{ns}$), we sample frames from the full trajectory at 20ns intervals. Starting from the native structure, the model learns to predict conformational changes over longer time scales, capturing major conformational transitions.

**High-resolution Training.** For the high-resolution model ($\Delta t_f = 1\text{ns}$), we randomly sample continuous trajectory segments of 20ns length with 1ns intervals. Each segment is an independent training sequence, allowing the model to learn local fluctuations and fast conformational changes.

**Notations:**

- $g_t \in \text{SE}(3)^L$: The frame sequence at time $t$, where each frame consists of rotation and translation components $(R, \mathbf{t})$ in SE(3) space
- $X_t \in \mathbb{R}^{L \times 13}$: Protein conformation at time $t$, where each residue includes 13 features as described in Section B.
- $T$: Number of timesteps used for both input and prediction windows
- $K$: Total number of frames in the training trajectory (differs between scales)
- $\sigma_{noise}$: Noise scale in our SDE formulation

Algorithm 2 details our hierarchical inference strategy for generating complete protein dynamics trajectories. During inference, we first use the low-resolution model $f_\theta^s$ to predict conformations at coarse timesteps ($\Delta t_s = 20\text{ns}$), which captures slow collective motions. These coarse predictions then serve as anchoring points for the high-resolution model $f_\phi^f$, which fills in the intermediate frames at fine timesteps ($\Delta t_f = 1\text{ns}$) to capture local fluctuations. This hierarchical approach ensures that the generated trajectories maintain consistency between collective motions and fast local dynamics.

**Algorithm 1:** Autoregressive Training of Protein Dynamics Model with Multiple Timesteps

---

**Input:** ground truth frame sequences $\{g_t\}_{t=0}^{K-1} \in (\mathrm{SE}(3)^L)^K$,
ground truth trajectories $\{X_t\}_{t=0}^{K-1} \in (\mathbb{R}^{L\times 13})^K$,
amino acid identities $A \in \{1, \ldots, 20\}^L$,
teaching force probability $p_{tf}$,
number of timesteps $T$ (for both input and prediction)
**Output:** trained model parameters $\theta$

**1 for** *each training iteration* **do**
    // Initialize input buffers with first T frames
**2**    Initialize $X_{buffer} \leftarrow \{X_i\}_{i=0}^{T-1}$ ;
**3**    Initialize $g_{buffer} \leftarrow \{g_i\}_{i=0}^{T-1}$ ;
**4**    $\mathcal{L}_{total} \leftarrow 0$ ;
**5**    **for** $t = 0$ **to** $\lfloor K/T \rfloor - 2$ **do**
**6**        $\sigma_{noise} \sim \mathcal{U}(0.01, 0.05)$ ;
**7**        $\epsilon \sim \mathcal{N}(0, I)$ ;
**8**        $h_{latent} \leftarrow W_{proj}(X_{buffer} + \sigma_{noise}\epsilon)$ ;
**9**        $s \leftarrow \mathrm{Embed}(A)$ ;
**10**        **for** $i = 1$ **to** $n_{ipa}$ **do**
**11**            $h_{frame} \leftarrow \mathrm{InvariantPointAttention}(s, g_{buffer}, \sigma_{noise})$ ;
        // spatial processing
**12**        $h_{spatial} \leftarrow h_{latent} + h_{frame} + \mathrm{Embed}(\sigma_{noise})$ ;
**13**        **for** $j = 1$ **to** $n_{att}$ **do**
**14**            $h_{spatial} \leftarrow \mathrm{MultiHeadAttention}(h_{spatial})$ ;
        // temporal processing
**15**        $h_{temporal} \leftarrow h_{spatial}$ ;
**16**        **for** $k = 1$ **to** $n_{GRU}$ **do**
**17**            $h_{temporal} \leftarrow \mathrm{GRU}(h_{temporal})$ ;
**18**        $X_{pred} \leftarrow W_{proj}(h_{temporal})$ ;
**19**        $g_{pred} \leftarrow \mathrm{RigidTransformDecode}(X_{pred})$ ;
**20**        $\mathcal{L}_{mse} \leftarrow \|X_{pred} - X_{(t+1)T:(t+2)T}\|^2$ ;
**21**        $\mathcal{L}_{clash} \leftarrow \mathrm{ComputeClashScore}(X_{pred})$ ;
**22**        $\mathcal{L}_{total} \leftarrow \mathcal{L}_{total} + \mathcal{L}_{mse} + \lambda\mathcal{L}_{clash}$ ;
        // Update buffers
**23**        $r \sim \mathcal{U}(0, 1)$ ;
**24**        **if** $r > p_{tf}$ **then**
**25**            $X_{buffer} \leftarrow X_{pred}$ ;
**26**            $g_{buffer} \leftarrow g_{pred}$ ;
**27**        **else**
**28**            $X_{buffer} \leftarrow X_{(t+1)T:(t+2)T}$ ;
**29**            $g_{buffer} \leftarrow g_{(t+1)T:(t+2)T}$ ;
**30**    Update model parameters $\theta$ using $\nabla_\theta \mathcal{L}_{total}$ ;

---

# D  Evaluation Metrics

We employ a comprehensive set of metrics to evaluate both structural accuracy and dynamic properties of generated protein ensembles. Our evaluation framework follows established protocols in protein ensemble generation [19], adapted for trajectory-based assessment.

## D.1  Structural Flexibility Metrics

**Pairwise RMSD.** For each ensemble, we quantify overall conformational diversity as the average C$\alpha$-RMSD between all pairs of conformations. This metric captures the range of conformational space explored by the ensemble. We report both the absolute values and Pearson correlation coefficient

---

**Algorithm 2:** Multi-scale Inference for Protein Dynamics Generation

---

**Input:** initial frames $\{X_i\}_{i=0}^{T-1}, \{g_i\}_{i=0}^{T-1}, ;$      `// T frames as initial buffer`
   **1** coarse timestep $\Delta t_s = 20\text{ns}$,
   **2** fine timestep $\Delta t_f = 1\text{ns}$,
   **3** total simulation time $T_{total}$,
   **4** trained coarse-resolution model $f_\theta^s$,
   **5** trained fine-resolution model $f_\phi^f$

**Output:** complete trajectory $\mathcal{X} = \{X_t\}_{t=0}^{T_{total}-1}$

**6** $X_{buffer}^s \leftarrow \{X_i\}_{i=0}^{T-1}$;
**7** $g_{buffer}^s \leftarrow \{g_i\}_{i=0}^{T-1}$;
**8** $\mathcal{X} \leftarrow \{X_i\}_{i=0}^{T-1}$;
**9** **for** $t = 0$ **to** $n_s$ **do**
**10**     $X_{pred}^s, g_{pred}^s \leftarrow f_\theta^s(X_{buffer}^s, g_{buffer}^s)$;
**11**     $X_{buffer}^f \leftarrow X_{pred}^s$;
**12**     $g_{buffer}^f \leftarrow g_{pred}^s$;
**13**     Append $X_{pred}^s$ to $\mathcal{X}$;
**14**     **for** $k = 0$ **to** $n_f$ **do**
**15**        $X_{pred}^f, g_{pred}^f \leftarrow f_\phi^f(X_{buffer}^f, g_{buffer}^f)$ `// Model forward pass`
**16**        Append $X_{pred}^f$ to $\mathcal{X}$;
**17**        Update $X_{buffer}^f, g_{buffer}^f$ with prediction;
**18**     Update $X_{buffer}^s, g_{buffer}^s$ with prediction;
**19** **return** $\mathcal{X}$

---

between predicted and ground truth pairwise RMSD distributions to assess whether our model captures the relative flexibility patterns across different proteins.

**Root Mean Square Fluctuation (RMSF).** To assess local flexibility, we compute the RMSF for each residue, measuring the standard deviation of atomic positions across the ensemble after optimal alignment. The Pearson correlation between predicted and ground truth RMSF profiles indicates how well the model captures residue-level flexibility patterns.

### D.2 Distribution Accuracy Metrics

**Root Mean Wasserstein Distance (RMWD).** To generalize all-atom RMSD to ensemble comparison, we define the root mean Wasserstein distance between ensembles $\mathcal{X}$ and $\mathcal{Y}$ as:

$$\text{RMWD}(\mathcal{X}, \mathcal{Y}) = \sqrt{\frac{1}{N} \sum_{i=1}^{N} \mathcal{W}_2^2 \left( \mathcal{N}[\mathcal{X}_i], \mathcal{N}[\mathcal{Y}_i] \right)} \tag{22}$$

where $\mathcal{N}[\mathcal{X}_i]$ denotes a 3D Gaussian fitted to the positional distribution of the $i$th atom in ensemble $\mathcal{X}$. This metric reduces to standard RMSD for single structures and provides a distributional measure of positional accuracy.

**Principal Component Analysis.** To evaluate collective motions, we project the joint distribution of C$\alpha$ positions onto principal components computed from the MD ensemble. We measure: (1) the 2-Wasserstein distance between predicted and ground truth ensembles in the PC space (MD PCA W2), and (2) the cosine similarity between the dominant principal components. A similarity $> 0.5$ indicates successful capture of the dominant collective motion. We report the percentage of test proteins achieving this threshold (% PC-sim $> 0.5$).

### D.3 Dynamic Property Metrics

**Contact Dynamics.** We analyze intermittent contacts to assess if the model captures thermal fluctuations. For each ensemble, we identify: (1) *weak contacts* - C$\alpha$ pairs ($<8$Å) in the native

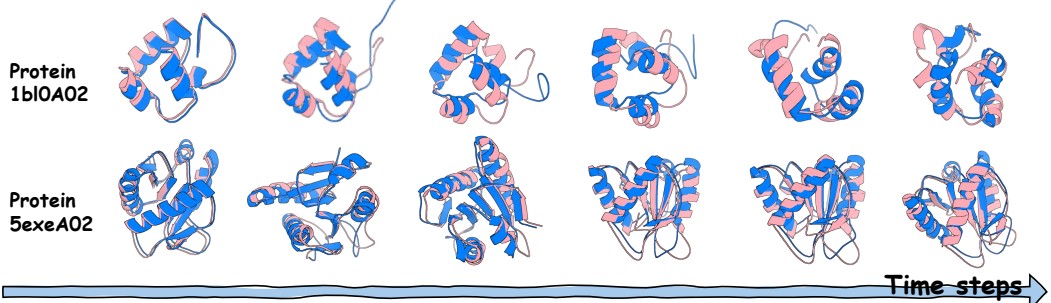

Figure 6: Visualization of protein conformational changes over time for two representative proteins (1bl0A02 and 5exeA02). Each row shows the temporal evolution of one protein. MD ground truth trajectories are shown in blue, while TEMPO-generated trajectories are shown in pink, demonstrating the close structural alignment between generated and reference structures throughout the trajectory.

structure that dissociate in $> 10\%$ of ensemble structures, and (2) *transient contacts* - pairs not in contact in the native structure but associate in $> 10\%$ of ensemble structures. We compute Jaccard similarity between predicted and ground truth contact sets.

**Trajectory Accuracy.** For trajectory generation methods, we measure the backbone RMSD error: $\text{Error}_{\text{frame}} = |\text{RMSD}_{\text{pred}} - \text{RMSD}_{\text{gt}}|$, where RMSD is computed relative to the native structure. This metric quantifies the model's ability to accurately capture the magnitude of conformational changes over time.

**Clash Ratio.** We compute the proportion of generated conformations containing steric clashes, defined as backbone atom pairs from different residues with distance $< 1.2\text{Å}$. This metric validates the physical plausibility of generated structures.

All metrics are computed on backbone atoms (N, C$\alpha$, C, O) after optimal rigid-body alignment to the native structure, unless otherwise specified. We report median values across the test set to ensure robustness to outliers.

# E    Trajectory Visualization

To provide an intuitive visualization of our generation results, we present trajectory comparisons for two representative test proteins. Figure 6 shows snapshots of the trajectories at $80\text{ns}$ intervals, where the ground truth MD conformations are shown in blue and TEMPO-generated conformations are shown in pink. These visualizations demonstrate the close structural alignment between our generated conformations and the MD reference states.

# F    Thermodynamic Accuracy Analysis

Following BioEMU's established evaluation methodology [23], we provide quantitative assessment of thermodynamic accuracy through free energy difference analysis. This analysis validates that our generated trajectories not only capture structural features but also preserve the underlying thermodynamic properties of protein dynamics.

**Free Energy Difference Computation.** We compute free energy differences by extracting reaction coordinates, specifically the fraction of native contacts, to calculate folding probabilities ($p_{\text{fold}}$). The free energy differences are then calculated as:

$$\Delta G = -kT \cdot \log\left(\frac{p_{\text{fold}}}{1 - p_{\text{fold}}}\right) \tag{23}$$

where $k$ is the Boltzmann constant and $T$ is the temperature. We measure the free energy error as $\Delta\Delta G = \Delta G_{\text{ground truth}} - \Delta G_{\text{predicted}}$.

**Results.** TEMPO achieves an average $\Delta\Delta G$ of 0.67 kcal/mol on the mdCATH test set, demonstrating good agreement with reference MD simulations within the acceptable range for biological applications

(typically <1-2 kcal/mol). This result indicates that our method not only generates structurally accurate conformations but also preserves the thermodynamic properties of protein folding and unfolding processes.

**Free Energy Profiles.** Following BioEMU's evaluation framework, we analyze our generated trajectories through multiple perspectives:

- **1D Free Energy Profiles:** We construct free energy profiles using three key reaction coordinates: RMSD from native structure, radius of gyration, and fraction of native contacts. Our generated trajectories exhibit high similarity to ground truth MD simulations across all three coordinates.

- **2D Free Energy Surfaces:** Two-dimensional free energy surface plots constructed from combinations of reaction coordinates show that TEMPO captures the essential features of the conformational landscape, including energy minima locations and barrier heights.

- **Time Series Analysis:** The fraction of native contacts time series from our generated trajectories closely matches the temporal evolution patterns observed in ground truth simulations, validating our model's ability to capture dynamic processes.

These quantitative thermodynamic evaluations complement our structural metrics and demonstrate that TEMPO generates trajectories that are not only geometrically accurate but also thermodynamically consistent with reference MD simulations (see Figure 7 and Figure 8 for visualizations).

# G   Ablation Study: Multi-scale vs Single-scale Generation

While autoregressive methods typically suffer from error accumulation during sequential generation, our hierarchical multi-scale design mitigates this issue by anchoring fine-scale dynamics to coarse-scale predictions. To demonstrate the effectiveness of our multi-scale architecture, we compare TEMPO's full hierarchical framework against a single-scale baseline that directly generates 400 frames without multi-scale guidance on mdCATH.

Table 3 shows that the multi-scale approach significantly outperforms single-scale generation across all metrics. The multi-scale model achieves substantially lower RMSD error (1.78Å vs 8.62Å) after 400 frames, demonstrating controlled error propagation that maintains trajectory stability over extended generation periods. Furthermore, the multi-scale design preserves conformational diversity (pairwise RMSD of 2.78Å vs 7.46Å compared to ground truth 3.26Å) and structural flexibility patterns (Pearson correlation of 0.77 vs 0.14 for RMSD, 0.67 vs 0.15 for RMSF), while the single-scale approach shows severe degradation in capturing protein dynamics. These results validate that our hierarchical decomposition is crucial for generating physically realistic long protein trajectories.

Table 3: Ablation study comparing multi-scale TEMPO with single-scale baseline on mdCATH. Ground truth values are shown in parentheses where applicable. The single-scale model directly generates 400 frames without hierarchical guidance.

| Metrics | TEMPO (Multi-scale) | TEMPO (Single-scale) |
|---|---|---|
| Pairwise RMSD (= 3.26) | **2.78** | 7.46 |
| Pairwise RMSD $r \uparrow$ | **0.77** | 0.14 |
| All-atom RMSF (= 1.64) | **1.60** | 4.27 |
| Global RMSF $r \uparrow$ | **0.67** | 0.15 |
| Root mean W2 $\downarrow$ | **4.21** | 8.27 |
| MD PCA W2 $\downarrow$ | **2.33** | 2.53 |
| % PC-sim $> 0.5 \uparrow$ | **7.81** | 3.12 |
| Weak contacts $J \uparrow$ | **0.43** | 0.23 |
| Trans. contacts $J \uparrow$ | **0.20** | 0.09 |
| RMSD Error $\downarrow$ | **1.78** | 8.62 |

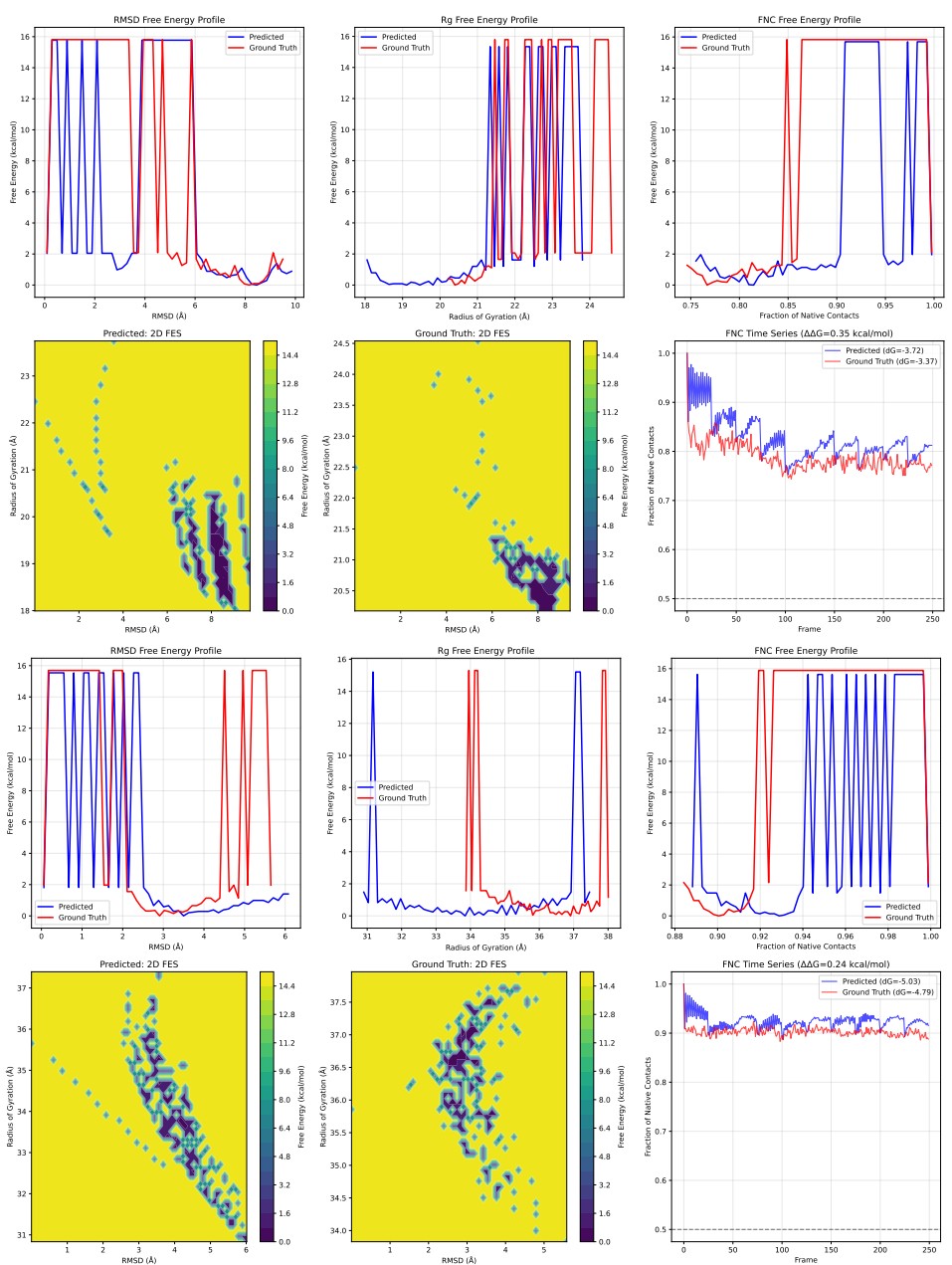

Figure 7: Thermodynamic accuracy analysis for two representative proteins (7p46_A and 7asg_A) selected from the ATLAS test set. For each protein, we show: 1D free energy profiles along three reaction coordinates (RMSD, radius of gyration, and fraction of native contacts), 2D free energy surfaces, and time series of fraction of native contacts.

# H Training vs Test Performance Analysis

We acknowledge that capturing all distribution modes represents a fundamental challenge in protein dynamics modeling. While our primary objective focuses on generating realistic conformational transitions rather than perfect mode coverage, we recognize the importance of understanding the performance gap between training and test scenarios.

**Performance Gap Analysis.** Table 4 compares TEMPO's performance on training and test sets of mdCATH, where the training set results are evaluated on 60 randomly selected proteins from the

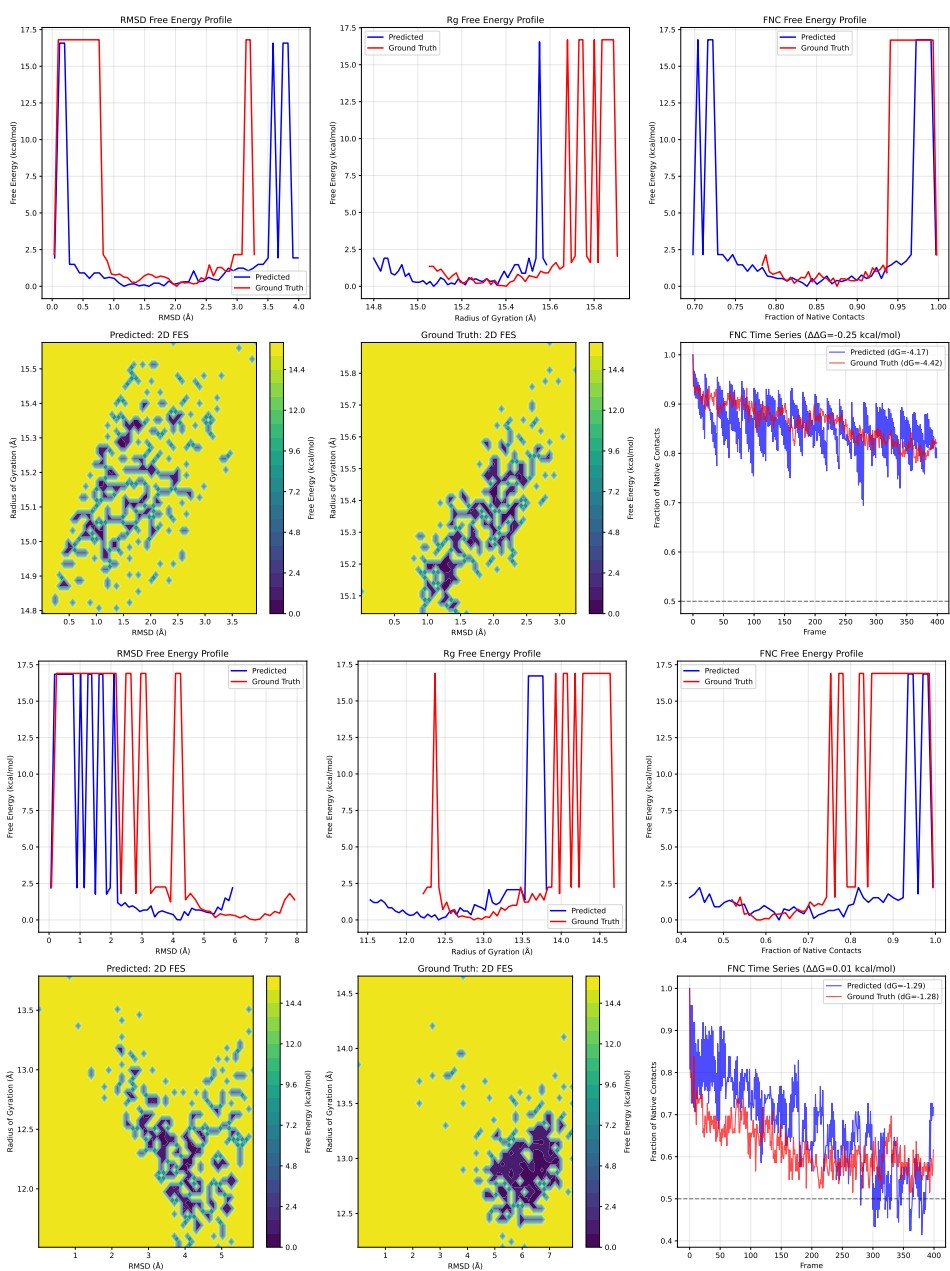

Figure 8: Thermodynamic accuracy analysis for two representative proteins (2ijd101 and 1rl0A02) selected from the mdCATH test set.

training data. The training set demonstrates significantly better mode coverage accuracy, with 46.7% of principal components achieving similarity greater than 0.5 compared to 7.81% on the test set. Similarly, the training set shows improved performance across most metrics, including lower MD PCA Wasserstein distance (1.31 vs 2.33) and better contact prediction accuracy (0.53 vs 0.43 for weak contacts, 0.37 vs 0.20 for transient contacts).

This performance gap reflects the inherent complexity of generalizing to novel protein architectures in mdCATH, where most proteins contain distinct energy basins and unique conformational landscapes. The challenge of mode coverage is not unique to our method - even extensively pretrained methods like BioEMU show significant free energy surface (FES) deviations as shown in Appendix Figure 11. The difficulty stems from the fundamental nature of protein dynamics: each protein has its own

characteristic energy landscape shaped by its unique sequence and structure, making it challenging for models to perfectly capture the full conformational distribution of unseen proteins.

Despite the performance gap, our test set results still demonstrate strong capability in capturing essential protein dynamics, as evidenced by high structural flexibility correlations (Pearson $r = 0.77$ for pairwise RMSD, $r = 0.67$ for RMSF) and accurate trajectory generation (RMSD error of 1.78Å). The stronger training set performance validates our model's capacity to learn complex protein dynamics when sufficient data is available, suggesting that expanded training data incorporating diverse protein architectures could further improve generalization.

Table 4: Comparison of TEMPO performance on training set versus test set. Ground truth values are shown in parentheses. The training set demonstrates significantly better mode coverage.

| Metrics | Train Sample | Test |
|---|---|---|
| Pairwise RMSD | 3.06 (2.51) | 2.78 (3.26) |
| Pairwise RMSD $r$ ↑ | **0.79** | 0.77 |
| All-atom RMSF | 1.16 (1.31) | 1.60 (1.64) |
| Global RMSF $r$ ↑ | **0.78** | 0.67 |
| Root mean W2 ↓ | **3.42** | 4.21 |
| MD PCA W2 ↓ | **1.31** | 2.33 |
| % PC-sim $> 0.5$ ↑ | **46.7** | 7.81 |
| Weak contacts $J$ ↑ | **0.53** | 0.43 |
| Trans. contacts $J$ ↑ | **0.37** | 0.20 |
| RMSD Error ↓ | **1.51** | 1.78 |

## I Additional Up-sampling Analysis

Additional up-sampling experiments across multiple test proteins further validate our high-resolution model's ability to generate full protein dynamics from ground truth low-resolution protein conformations. Figure 9 shows the FES contour plots for four randomly selected proteins (1rl0A02, 1x4tA01, 1zpdA02, and 2ndpA00) computed from their backbone conformations.

## J Additional State Transition Analysis

To further validate our model's capability in capturing protein conformational transitions, we present additional state transition analyses on representative proteins from our test set. Figure 10, each subplot compares the conformational trajectories generated by TEMPO and MDGen with MD simulations in the space of the first two principal components. Consistent with our observations in the main text, these additional examples demonstrate TEMPO's robust ability to generate physically meaningful transition pathways. MDGen typically exhibits more clustered sampling patterns, fails to capture the continuous nature of conformational transitions. These results further support our framework's advantage in modeling the temporal dependencies inherent in protein dynamics.

## K Additional Free Energy Surface Analysis

To complement the FES analysis, we randomly selected four additional proteins from our test set for detailed comparison. As shown in Figure 11. Based on the FES analysis across four randomly selected test proteins, we observe several distinctive patterns in conformational sampling strategies: TEMPO demonstrates precise conformational sampling that closely aligns with the ground truth distributions. The generated conformations (orange points) are concentrated within physically meaningful energy basins, suggesting coherent trajectory generation. MDGen shows similar sampling quality, though with slightly more dispersed distributions in some cases.

In contrast, AlphaFlow exhibits broader but less focused sampling, often deviating from the main energy wells. BioEMU shows the most scattered sampling patterns, frequently generating physically implausible conformations that lie outside the main energy basins. This comparison highlights the advantage of our trajectory-aware approach over independent sampling methods - while methods like

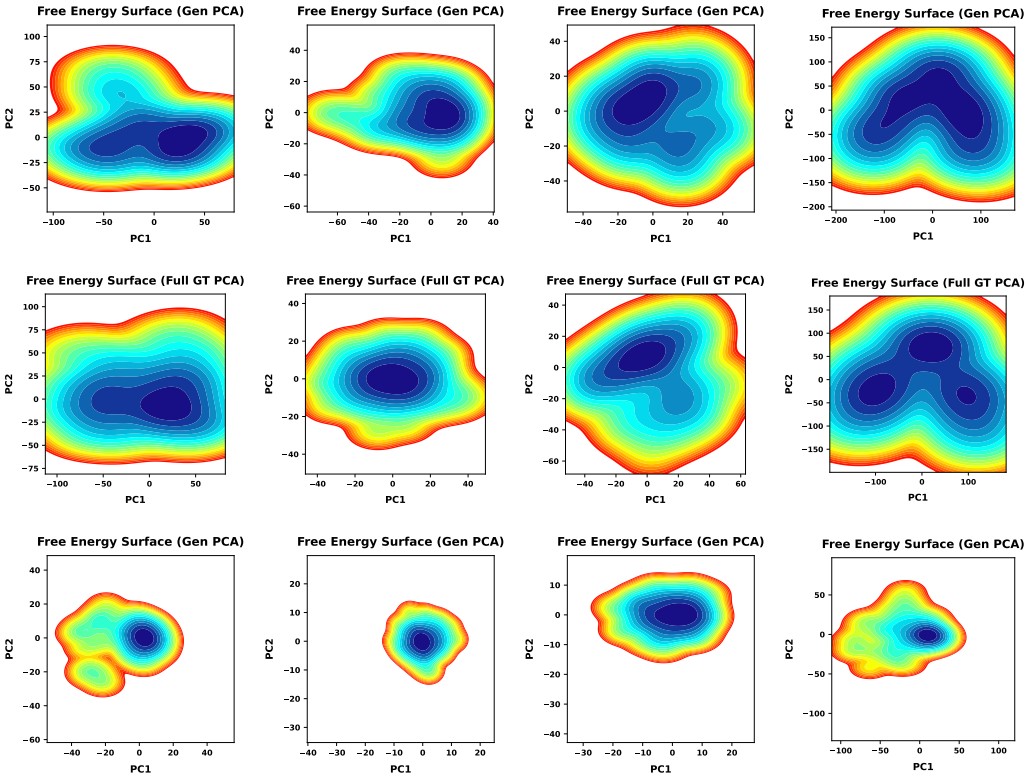

Figure 9: FES plots of four randomly selected test proteins. The FES is computed from: (top) trajectories generated by TEMPO performing up-sampling tasks, (middle) ground truth molecular dynamics simulations, and (bottom) trajectories produced by the baseline model MDGen, performing simulation tasks. Each column represents a distinct test protein (from left to right: 1rl0A02, 1x4tA01, 1zpdA02, and 2ndpA00).

AlphaFlow and BioEMU may achieve wider conformational coverage, they often do so at the cost of physical realism.

These observations consistently demonstrate that TEMPO's multi-scale framework effectively balances conformational exploration with physical constraints, producing trajectories that maintain both continuity and thermodynamic plausibility. The results validate our design choice of incorporating temporal dependencies, which proves crucial for generating biologically meaningful protein dynamics.

## L   Per-Residue RMSF.

The RMSF analysis reveals our model's ability to capture local protein dynamics across different scales. Figure 12 illustrates a representative case study using protein 4impA02, where the generated trajectories closely mirror the MD simulation's $C\alpha$ fluctuation patterns. In this example, both profiles exhibit characteristic mobility signatures, with enhanced fluctuations at the N-terminus (residues 0-20) and C-terminus (residues 190-210), while maintaining relatively stable conformations (RMSF < 2Å) in the central regions. Across our entire test set, the generated trajectories maintain a reasonable correlation (average Pearson r = 0.67) with MD simulations in terms of RMSF profiles, suggesting that our model consistently reproduces biologically relevant flexibility patterns. This global performance indicates that the model has learned meaningful protein dynamics patterns rather than generating arbitrary motions. Additional case studies are provided in Figure 13.

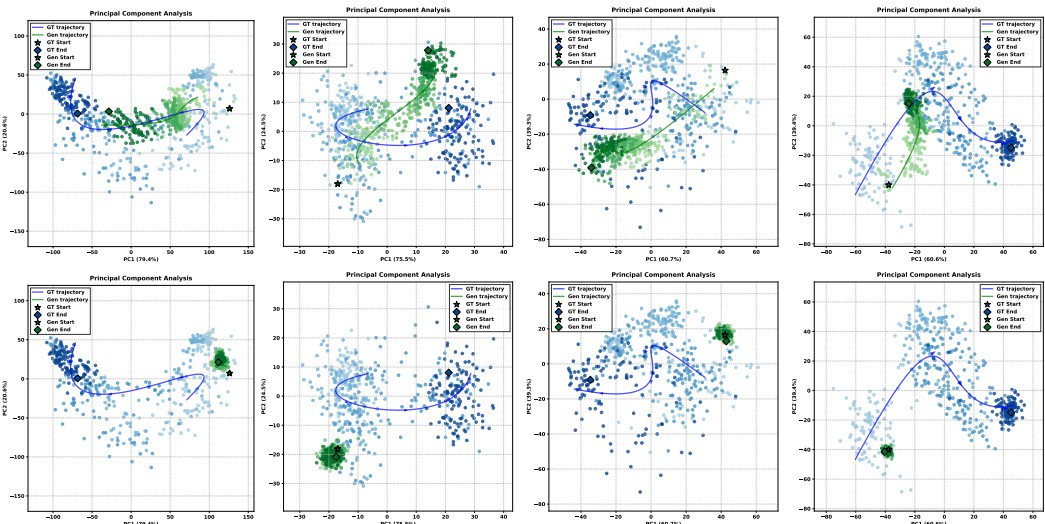

Figure 10: Comparison of conformational transitions in PC space between TEMPO and MDGen baseline (bottom). Ground truth MD trajectories are shown in blue, while generated trajectories are in green. The polynomial fitting curves highlight the temporal evolution of conformational changes.

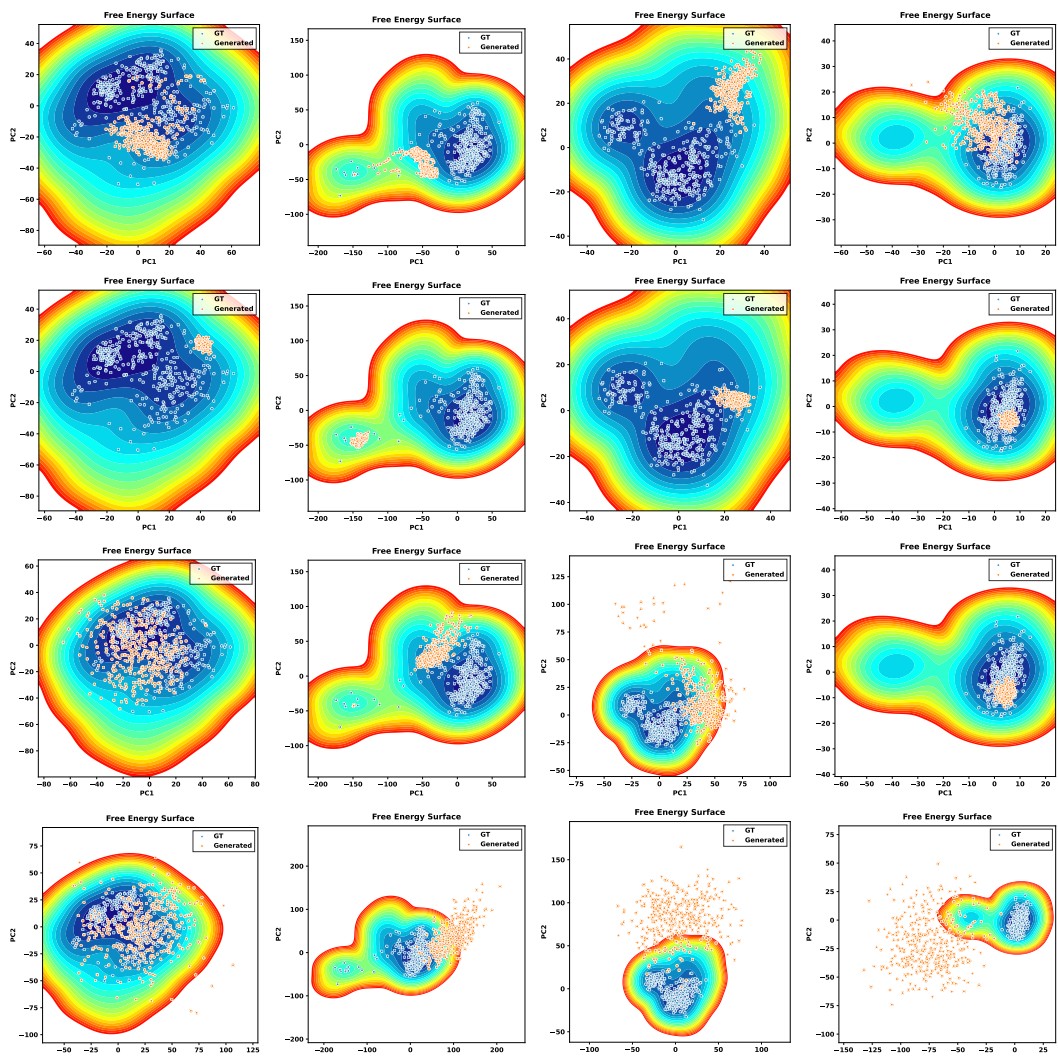

Figure 11: Comparison of FES for four randomly selected test proteins (from left to right: 3cj8B02, 3cx5E01, 3f6kA03, 4fomA03). Results are shown for TEMPO, MDGen, AlphaFlow, and BioEMU (from top to bottom).

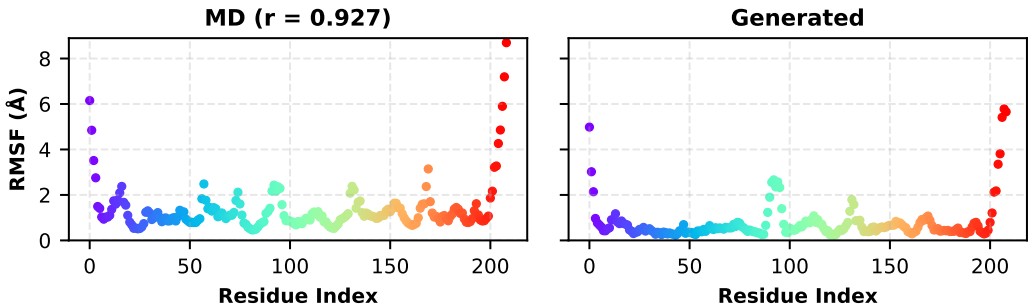

Figure 12: Comparison of Root Mean Square Fluctuation (RMSF) between MD simulation trajectory and generated trajectory for protein 4impA02. The RMSF values reflect the C$\alpha$ fluctuations of protein residues during the simulation. The Pearson (r) between the two RMSFs is $0.93$.

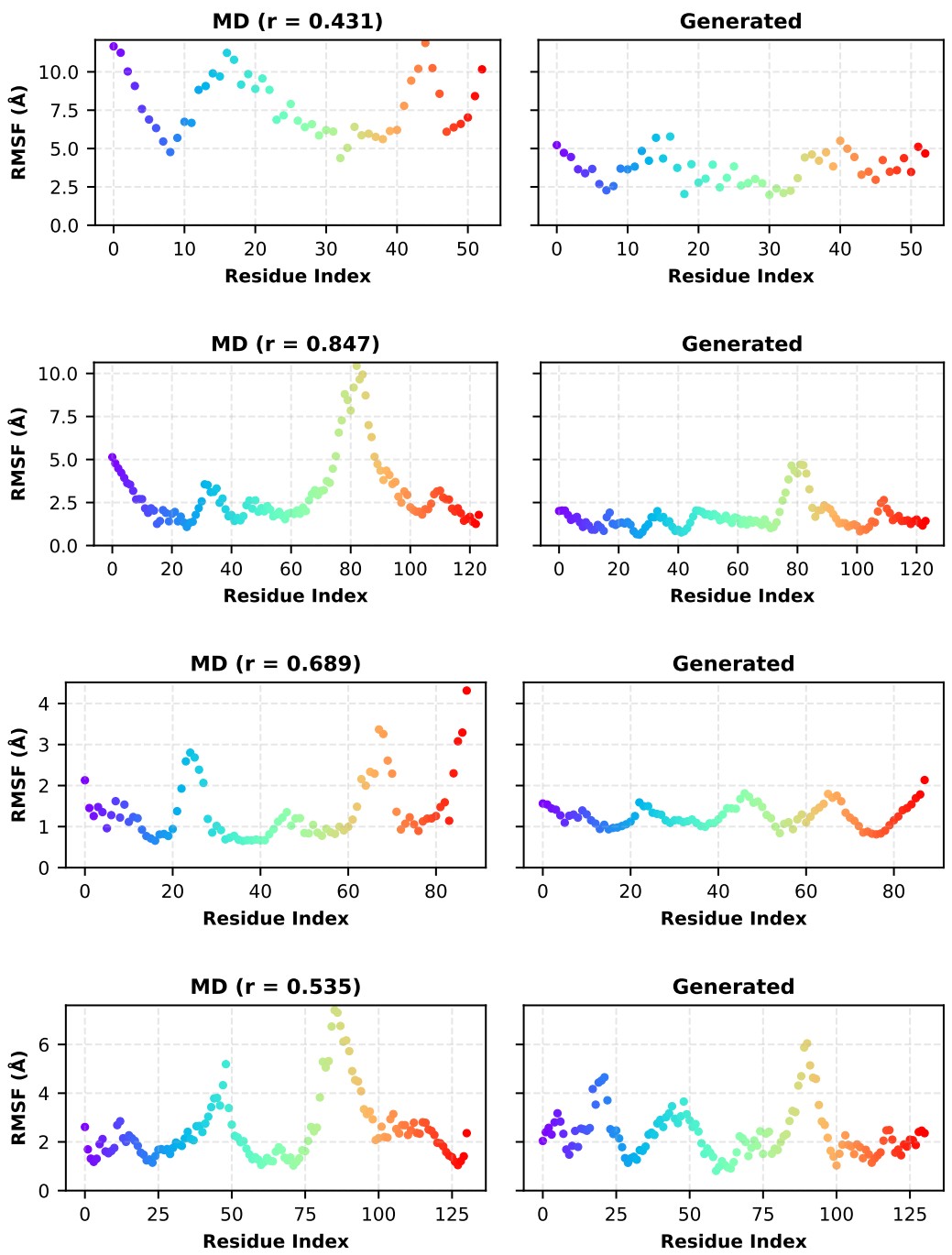

Figure 13: RMSF profiles comparison between MD simulations and TEMPO generated trajectories for 4 randomly selected test proteins. Each plot demonstrates the Cα atomic fluctuations along the protein sequence (Protein from top to bottom: 1y4mA00, 3b8xA02,3bjdA01, 3gyxA02).

