# OpenReview forum: "TEMPO: Temporal Multi-scale Autoregressive Generation of Protein Conformational Ensembles"
_NeurIPS.cc/2025/Conference — NeurIPS 2025 poster_

### Official Review · Reviewer_HVNL · 2025-07-01

**Clarity:** 3
**Significance:** 2
**Originality:** 2
**Rating:** 4
**Confidence:** 4

**Summary:**

In this paper, the authors present TEMPO, a method to generate trajectories of protein dynamics by learning a multi-scale dynamical propagator parameterized by a neural network. Specifically, capitalizing on the multi-scale nature of protein dynamics, the authors propose to learn a coarse propagator which evolves the system at a large timestep (e.g. 20 ns) to capture large-scale conformational motions, as well as a fine propagator which refines the coarse prediction at a smaller timestep (e.g. 1 ns). Results are demonstrated on proteins from the mdCATH dataset, showing that TEMPO can recover conformational ensembles and transition pathways of proteins well relative to baseline methods.

**Questions:**

1. Is there a reason the authors chose to frame the prediction task as a deterministic mapping from $X_t$ to $X_{t+\Delta t}$ and adding Langevin noise after the prediction, as opposed to treating the problem in a generative modeling/distribution matching/stochastic interpolant framework (i.e sample from the distribution $p(X_{t+\Delta t} | X_t)$). It seems that that would make more sense, because the ground truth $(X_t$ to $X_{t+\Delta t})$ tuples that the method is being trained on are presumably the result of stochastic dynamics, so it would be incorrect to treat this $X_{t+\Delta t}$ as the unique ground truth value. Additionally, this way it would not be necessary to set/sample an empirical noise scale during sampling.

2. Does switching from the fine to the coarse dynamics every e.g. 20 steps introduce boundary effects/discontinuities in the sampled dynamics?

3. During training, through how many autoregressive steps are you able to optimize the loss function without exhausting memory/getting unstable gradient norms? Within this limit, does reducing the teacher forcing probability and backpropagating though more autoregressive steps lead to performance improvements?

4. Just to clarify, were all baselines retrained on the identical training split of mdCATH?

**Ethical Concerns:**

["NO or VERY MINOR ethics concerns only"]

**Final Justification:**

I thank the authors for addressing some of my concerns during the rebuttal period, and have raised my score from 3 to 4.

In general, this paper proposes an interesting approach for generating temporal trajectories of protein dynamics by coupling fast and slow timescales. While this type of approach is not entirely novel and has a history in the molecular dynamics literature, to my knowledge it has not been combined with generative models in this way, and appears to offer promising speedups over baselines (although one point that was unclear to me after going over the inference times reported in Table 1 is whether the reported time for 400 snapshots is comparing correlated snapshots in the case of TEMPO/MDGen to uncorrelated snapshots for ESM/AlphaFlow - if this is the case, then this would be an unfair comparison, as the latter is sampling more statistically independent samples in the given time. I would request the authors to clarify this in their final version).

Overall, I recommend weak acceptance due to the fact that the ideas contained in the paper are interesting, but I still don't find the results/claims of superior sampling to be very convincing (e.g Figures 4, 8, 9). I think the paper would be strengthened more with clearer examples of how the generated samples are substantively more useful/accurate than the baselines - the visual results in the aforementioned figures are not convincing on their own.

**Limitations:**

Yes

**Quality:**

2

**Strengths And Weaknesses:**

Strengths:

1. The paper is generally well-written and easy to follow.

2. The proposed method of learning a fast and slow dynamical propagator is conceptually simple and easy to implement. It allows for the possibility of simply generating coarse dynamical trajectories without resolving the finer/faster dynamics, which is often sufficient for many practical applications.

Weaknesses:
1. It seems that the training procedure requires pre-specification of the integration timestep during training, and that this cannot be changed during inference. In contrast, previous works directly [1] learn the transition density operator, which allows for conditioning on an arbitrary timestep (and recovering i.i.d sampling as $\Delta t \rightarrow \infty $). In addition, the hierarchy is fixed to be 2-levels. This seems to limit the flexibility of the approach - there may be systems for which less or more temporal resolution is required.

2. The strength of the results appears inconsistent and unconvincing.

a) In Table 1, apart from the RMSD metrics which loosely capture whether the correct amount of conformational diversity is being captured, performance from TEMPO appears mixed/weak relative to baselines. In particular, the % clash ratio metric (which presumably measures the frequency of steric clashes in the produced dynamics) is considerably higher than MDGen, making it unclear whether the proposed steric clash loss is having the intended effect.

b) Additionally, it appears from Figure 2 that the dynamics learned by TEMPO are highly smoothed compared to the reference. Is this because of the 1 ns fine timescale?

c) In Figures 4 and 5, it is not at all clear to me that TEMPO-produced trajectories are superior to those of the baselines. In most cases, the sampled points do not even cover all the modes of the distribution, suggesting that large-scale conformational transitions are not being captured accurately. Is this perhaps a generalization issue (i.e are results better on the training proteins but fail to translate to the test set)?

d) The claim that TEMPO produces more "focused exploration" of the conformational space than the baselines is at odds with the claim that it enables sampling large-scale conformational transitions.

[1] Schreiner, et. al "Implicit Transfer Operator Learning: Multiple Time-Resolution Surrogates for Molecular Dynamics"

---

> ### Author Rebuttal · Authors · 2025-07-31
>
> We sincerely thank the reviewer for their detailed feedback on our modeling framework choices, evaluation methodology, and concerns about result consistency and generalization performance. Below are our detailed responses addressing each point.
>
> **W1**
>
> We appreciate this comment. While our current implementation uses pre-specified timesteps, this design choice reflects the established biophysical principle that protein dynamics naturally separate into slow collective motions and fast local fluctuations. The framework can be naturally extended to multiple scales by adding coupled SDE terms, similar to adaptive resolution schemes in image generation.
> Unlike ITO, which focuses on coarse-grained Cα representations and uses exponential distribution sampling, our approach models full backbone dynamics while maintaining temporal coherence and biological interpretability. ITO shows challenges in long-time scale prediction (as evidenced by negative VAMP-2 gaps that increase with lag time). In contrast, our hierarchical approach decomposes complex long-timescale dynamics into more tractable multi-step processes. Additionally, ITO was only evaluated on 12 folding proteins, making it difficult to assess its model capacity and generalization ability on larger protein datasets, whereas our model is trained on the comprehensive mdCATH dataset and tested on novel proteins to demonstrate robust generalization capabilities. We will provide the discussion of these methodological differences in the revision.
>
>
> **W2.a**
>
> We appreciate this observation and note that unlike ensemble methods that perform independent sampling, TEMPO generates temporally coherent MD trajectories while maintaining thermodynamically consistent free energy surfaces, demonstrating comparable ensemble quality and superior trajectory fidelity through lower RMSD error compared to MDGEN, and while our clash ratio is higher than MDGEN, it remains significantly lower than ensemble-based baselines like BioEMU and AlphaFlow, with visualizations revealing that MDGEN‘'s lower clash ratio comes at the cost of severely restricted sampling confined to very small conformational regions on the mdCATH dataset, and our clash loss has already substantially improved inference stability with future incorporation of enhanced physical constraints expected to further reduce the clash ratio.
>
> **W2.b**
>
> The smoothed curve in Figure 2 occurs because we compare our coarse-resolution trajectories (20ns intervals) with full MD ground truth (1ns intervals), as our coarse model is designed to capture slow collective motions rather than high-frequency fluctuations.
>
> **W2.c**
>
> We thank the reviewer for raising this concern about mode coverage challenges and acknowledge that capturing all distribution modes represents a fundamental difficulty in protein dynamics modeling, as even extensively pretrained methods like BioEMU show significant FES deviations (Appendix Figure 9). While our primary objective focuses on generating realistic conformational transitions rather than perfect mode coverage (Figure 4 and Appendix Figure 8), we recognize the performance gap between training and test scenarios reflects the complexity of generalizing to novel protein architectures in mdCATH where most proteins contain distinct energy basins. We have added TEMPO's performance on the training set in the table below, evaluated on 60 randomly selected proteins, where the training set demonstrates significantly better mode coverage accuracy. Due to conference policy restrictions on supplementary figures, we will provide comprehensive training set visualizations in the revision to illustrate this performance.
>
> | Metrics | TEMPO (train sample) | TEMPO (test) |
> |---------|---------------------|----------------------|
> | Pairwise RMSD | 3.06(2.51) | 2.78(3.26) |
> | Pairwise RMSD r ↑ | **0.79** | 0.77 |
> | All-atom RMSF | 1.16(1.31) | 1.60(1.64) |
> | Global RMSF r | **0.78** | 0.67 |
> | Root mean W2 ↓ | **3.42** | 4.21 |
> | MD PCA W2 ↓ | **1.31** | 2.33 |
> | % PC-sim > 0.5 ↑ | **46.7** | 7.81 |
> | Weak contacts J ↑ | **0.53** | 0.43 |
> | Trans. contacts J ↑ | **0.37** | 0.20 |
> | RMSD Error ↓ | **1.51** | 1.78 |
>
> **W2.d**
>
> We clarify that "focused exploration" refers to TEMPO's adaptive sampling around physically relevant conformational regions rather than uniform dispersion, which enables more efficient capture of large-scale transitions when they occur (Figure 4 and Figure 8 in the Appendix). This contrasts with baseline methods that show more dispersed but less physically meaningful patterns, particularly evident in Figure 9 in the Appendix, where our trajectory-aware approach maintains tighter distributions around energetically favorable states while still capturing the essential conformational transitions that define protein dynamics.
>
> We appreciate the reviewer's feedback and respectfully note that protein dynamics generation represents an emerging and challenging field where our work addresses the more complex task of capturing temporal conformational evolution rather than just equilibrium ensemble generation, and our comprehensive evaluation demonstrates consistent improvements over baselines across multiple metrics on the demanding mdCATH dataset, with additional experiments on the ATLAS dataset(kindly refer to our respose to Q4) showing our model's substantial performance advantages over baselines through both quantitative metrics and visualizations, confirming our model's advanced ability to capture conformational transition pathways, particularly for proteins with complex free energy landscapes featuring multiple basins, while our multi-scale SDE formulation significantly reduces computational demands during training and maintains high efficiency during inference compared to direct trajectory decoding approaches used in prior work.
>
> **Q1**
>
> We thank the reviewer for this question about our modeling framework choice. Our deterministic prediction approach is theoretically justified because we are essentially learning the conditional expectation E[X_{t+1}|X_t] in the finite time-step regime, which mirrors how numerical MD integrators operate with deterministic updates plus controlled stochastic components for temperature regulation[1,2]. Additionally, our approach parallels transformer-based language models, where deterministic training (cross-entropy loss) still enables diverse generation during inference - similarly, we preserve stochastic diversity via Langevin noise while learning conditional expectations through MSE loss. At the 1ns sampling interval used in mdCATH, consecutive frames exhibit strong deterministic correlations due to physical inertia, making our model appropriate for capturing dominant drift dynamics, and our SDE formulation dX_t = μ(X_t)dt + σdW_t directly mirrors actual MD implementation, where deterministic force calculations drive conformational changes and stochastic thermostats introduce thermal effects.
>
> [1] Effective stochastic dynamics on a protein folding energy landscape.
>
> [2] Memory effects in irreversible thermodynamics.
>
> **Q2**
>
> We appreciate this question regarding trajectory continuity, and while potential discontinuities could theoretically occur in our hierarchical approach, our evaluation demonstrates that such instances are infrequent and remain within acceptable error ranges as evidenced by our low RMSD error metrics (Table 1) and smooth trajectory coherence (Figure 4 and Figure 8 in the Appendix), and developing more sophisticated continuity constraints represents an important direction for future work.
>
> **Q3**
>
> During training, we can optimize through approximately 20 autoregressive steps corresponding to our trajectory length under the multi-scale setting without encountering memory or gradient stability issues, and while we have not systematically explored reducing teacher forcing probability or extending backpropagation through longer sequences, this represents an important direction for future investigation that could potentially improve long-term trajectory coherence.
>
> **Q4**
>
> We appreciate the reviewer's question. We chose mdCATH over the widely used MD dataset ATLAS because its free energy landscapes contain more distinct energy basins with clearer conformational transition pathways better suited for our dynamics modeling task. At the same time, we evaluated baselines using their provided checkpoints since BioEMU has already been pretrained on diverse MD data and PDB structures; ESMFlow and AlphaFlow were not designed for MD generation. MDGEN encounters severe memory limitations on mdCATH due to its full-trajectory attention mechanism (requiring batch size of 1 and still experiencing out-of-memory errors on 8 A100 GPUs when training 400-frame conformations), but to address fairness concerns, we also trained TEMPO on ATLAS following MDGen's splits and results in table below demonstrate our method's strong performance across different datasets, achieving **state-of-the-art** results on ATLAS, with FES visualizations of generated ensembles corroborating our quantitative results, comprehensive visualizations will be provided in the revision due to current supplementary figure limitations.
>
>
> | Metrics | BioEMU | AlphaFlow-MD | MDGEN | TEMPO |
> |---------|-------|-------------|----------|---------|
> | Pairwise RMSD r ↑ | -0.02 | 0.48 | 0.48 | **0.91** |
> | Global RMSF r | 0.09 | 0.60 | 0.50 | **0.89** |
> | Root mean W2 ↓ | 19.23 | 2.61 | 2.69 | **1.49** |
> | MD PCA W2 ↓ | 3.61 | 1.52 | 1.89 | **0.60** |
> | % PC-sim > 0.5 ↑ | 14 | 44 | 10 | **76** |
> | Weak contacts J ↑ | 0.26 | 0.51 | 0.62 | **0.74** |
> | Trans. contacts J ↑ | 0.06 | 0.29 | **0.41** | 0.38 |
> | RMSD Error ↓ | - | - | 3.20 | **1.83** |

---

> > ### Comment · Reviewer_HVNL · 2025-08-05
> > **Response**
> >
> > I thank the authors for their response. Several of my concerns have been addressed. One additional point:
> >
> > Regarding Figure 5, I'm still not convinced that the TEMPO-produced results are superior/more useful than the ESMFlow baseline. I understand that due to the trajectory modeling, the states are more "focused"/tightly clustered around local regions, but often, we do care about sampling a diverse set of conformations that aren't "stuck" in local minima. Can the authors make a stronger case for why the sampling behavior exhibited in the top row of Fig 5 is substantially preferred to the behavior of the bottom row?

---

> > > ### Author Response · Authors · 2025-08-05
> > >
> > > We appreciate this important question and would like to clarify our sampling behavior and modeling objectives.
> > >
> > > ### 1. Clarification on Figure 5
> > >
> > > We do not claim Figure 5 demonstrates superior ensemble generation. Rather, it illustrates the **fundamental difference** between dynamics-based learning (TEMPO) and i.i.d. sampling (ESMFlow). TEMPO's focused sampling reflects our **deliberate modeling of temporal correlations**, not a limitation.
> > >
> > > For comprehensive evaluation, please kindly refer to **Figure 9 in the appendix** showing that TEMPO's sampling better matches real MD trajectories compared to BioEMU and AlphaFlow. The second column protein clearly shows conformational transitions from left to right regions - biologically meaningful trajectories impossible with independent sampling. Similar advantages are evident in **Figures 4 and 8**.
> > >
> > > ### 2. The Rationale Behind Our Design Choice
> > >
> > > This focused sampling behavior stems from our different modeling objective. While we agree diverse sampling is valuable for certain applications, our **temporal correlation modeling is specifically designed to generate realistic protein dynamics and dynamic ensembles**. This distinction is crucial for understanding biological mechanisms.
> > >
> > > As noted in our introduction, **temporally correlated dynamic ensembles are essential for understanding enzyme catalysis, drug-binding pathways, and allosteric regulation**. Dynamic ensembles can reveal cryptic binding sites accessible only through specific conformational transitions.
> > >
> > > ### 3. Why This Approach Is Biologically Justified
> > >
> > > TEMPO's focused sampling naturally emerges from fundamental physical constraints governing real protein motion:
> > > - **Kinetic accessibility**: Proteins must respect energy barriers and cannot instantaneously jump between distant conformations
> > > - **Sequential transitions**: Biological processes depend on pathway-dependent conformational changes
> > > - **Physical realism**: Conservation of momentum and energy barriers make many thermodynamically accessible states kinetically forbidden on biological timescales
> > >
> > >
> > > Therefore, TEMPO's focused sampling is a **deliberate design** that captures the temporal and physical constraints of real protein dynamics. While recognizing diverse sampling's value for other applications, our approach fills a critical gap in generating **biologically realistic dynamic ensembles** that preserve the sequential nature of conformational changes essential for mechanistic understanding.

---

> > > ### Author Response · Authors · 2025-08-06
> > >
> > > Thank you for your insightful question and suggestions, which have been very helpful in refining our work. We hope our recent response has answered your question about Figure 5. If you have any remaining concerns or need further clarification on any aspect of our work, we would be happy to provide additional details. If our responses have addressed your concerns, would you consider re-evaluating our work's contribution?

---

> ### Author Response · Authors · 2025-08-09
>
> Dear Reviewer HVNL,
>
> With the final hours of the discussion period approaching, we appreciate your engagement with our work. We're standing by if you have any additional questions or reflections.
>
> Thank you for your thoughtful comments.
>
> Best regards,
>
> The Authors

---

### Official Review · Reviewer_tNDp · 2025-07-02

**Clarity:** 3
**Significance:** 3
**Originality:** 3
**Rating:** 5
**Confidence:** 5

**Summary:**

This paper introduces TEMPO, a generative model for simulating protein dynamics through a multi-scale autoregressive architecture. It models both slow global motions and fast local fluctuations in protein conformations using SDEs and a two-level temporal model. They show the model can produce realistic, physically plausible, and temporally coherent trajectories.

The main contributions include:
1) They propose a multi-resolution temporal autoregressive framework for protein conformation generation.
2) They develop a spatiotemporal encoder that captures inter-residue geometry and temporal continuity.
3) The proposed method achieves strong experimental results and the sampling is efficient.

**Questions:**

1. The model seems to be trained from scratch. Have explored finetuning a casual-aware model from a pretrained structure prediction method?
2. How difficult would it be to extend TEMPO to model full-atom structures?
3. How do you choose the resolution hierarchy (variable $\Delta $ t) empirically?
4. Have you explored training a single resolution (say 1ns) model on full MD trajectories (in this case, 400 steps)?

**Ethical Concerns:**

["NO or VERY MINOR ethics concerns only"]

**Final Justification:**

Although some shortcomings remain, such as the fixed time hierarchy and potential inconsistencies in the generated trajectories, the proposed method provides a nice foundational framework for modeling protein conformational trajectories. The experiments are conducted on mdcath, a large-scale MD dataset that also deserves greater attention in the field. Overall, the strengths of the approach outweigh its limitations, and the combination of methodology and experimental validation makes me inclined to recommend acceptance of this paper.

**Limitations:**

yes

**Quality:**

3

**Strengths And Weaknesses:**

## Strengths

1. The decomposition of dynamics into slow and fast components looks interesting, which may reflect the biophysical principles.
2. Unlike many ensemble-focused methods, TEMPO ensures causal consistency across time via autoregressive sampling.
3. The proposed method achieve strong empirical results, while requiring fewer computational resources.
4. The paper is well written and they provide implementation details.

## Weakness
1. The two-scale time hierarchy is fixed to 20ns and 1ns, which might be suboptimal for arbitrary MD trajectories. Making the resolution learnable could improve the robustness of the algo.
2. Each high resolution trajectory fragment is initialized with the corresponding low resolution state, which might lead to inconsistent trajectories. The conformation may "jump" from one state to another during the fragment transition.
3. Currently the model only support backbone-only modeling, while sidechains are crucial for function and binding.
4. Unlinke  ensemble-based methods, the algorithms depend heavily on the quality and diversity of the training data (mdCATH dataset). It can not be extended to other cases (such as multimeric proteins) if we lack the data.
5. The algorithm takes the initial structure as input to generate the MD simulation and can not handle cases where initial structures are unavailable.

---

> ### Author Rebuttal · Authors · 2025-07-31
>
> We sincerely thank the reviewer for their thoughtful suggestions on adaptive timescale learning, trajectory continuity concerns, and potential extensions to full-atom modeling. Below are our detailed responses addressing each point.
>
> **W1**
>
> We appreciate this constructive suggestion and acknowledge that while our current fixed two-scale decomposition reflects the established biophysical separation of protein dynamics into slow collective motions and fast local fluctuations, our hierarchical SDE formulation can naturally be extended to learnable or adaptive timescales,  and we view this as a promising direction for future work that could enhance the framework's robustness across diverse protein systems with varying characteristic timescales.
>
> **W2**
>
> We appreciate this question regarding trajectory continuity, and while potential discontinuities could theoretically occur in our hierarchical approach, our evaluation demonstrates that such instances are infrequent and remain within acceptable error ranges as evidenced by our low RMSD error metrics (Table 1) and smooth trajectory coherence (Figure 4 and Figure 8 in Appendix), and developing more sophisticated continuity constraints represents an important direction for future work.
>
> **W3**
>
> We thank the reviewer for this valuable suggestion and clarify that, as an initial attempt, we focused on validating backbone trajectory generation within the TEMPO framework since backbone atoms determine the primary conformational changes of proteins, but we fully agree that sidechain generation is equally important and valuable. Extending to full-atom representation requires significantly larger model capacity and more physical constraints, making the incorporation of sidechain dynamics generation one of our key objectives for future model improvements.
>
> **W4**
>
> We appreciate this observation regarding data dependency, and note that while the current scarcity of comprehensive MD benchmarks and large-scale pretraining datasets for protein dynamics generation constrains our evaluation to mdCATH, our framework is designed to leverage temporal dependencies that become increasingly valuable as larger and more diverse MD datasets emerge, and we believe our approach will demonstrate advantages as the field develops richer benchmarks for protein dynamics modeling.
>
> **W5**
>
> The requirement for initial frames is inherent to the nature of molecular dynamics simulation and trajectory generation, as our method aims to generate temporally coherent MD trajectories that capture physically realistic transitions from a given starting conformation, and importantly, our model's performance is not limited by the source of the initial structure, allowing it to be coupled with any structure prediction method to generate dynamics from sequence-predicted conformations.
>
> **Q1**
>
> We appreciate this valuable suggestion and acknowledge that leveraging pretrained structure prediction models could potentially enhance our framework's performance, though we have not yet explored this direction due to computational constraints, representing an important avenue for future investigation.
>
> **Q2**
>
> Extending TEMPO to full-atom structures is conceptually straightforward, as our framework can accommodate higher-dimensional representations, though computational constraints currently limit our focus to backbone dynamics, which capture the essential conformational changes for most biological processes, and full-atom extension represents a key objective for our future model improvements.
>
> **Q3**
>
> We empirically chose the 20ns/1ns hierarchy based on established biophysical principles where the 20ns interval effectively captures major conformational transitions between different states in the free energy surface as visualized in Figure 1a, while the 1ns resolution represents the dataset's finest temporal sampling, and as a model hyperparameter, different datasets may require empirical exploration of various resolutions to optimize performance since protein dynamics timescales can vary significantly across different systems and functional contexts.
>
> **Q4**
>
> We have explored training single-resolution models at 1ns intervals and found that autoregressive error accumulation becomes problematic for extended trajectories, with single-scale generation beyond approximately 400 frames resulting in structural deviations exceeding 8Å RMSD relative to the native structure compared to ground truth. We also conducted ablation studies comparing our hierarchical TEMPO framework against single-scale baselines where the high-resolution model directly generates all frames without slow-scale guidance, as shown in the table below, demonstrating the importance of our multi-scale design.
>
> | Metrics | TEMPO (Multi-scale) | TEMPO (Single-scale) |
> |---------|---------------|-----------|
> | Pairwise RMSD(=3.26) | **2.78** | 7.46 |
> | Pairwise RMSD r ↑ | **0.77** | 0.14 |
> | All-atom RMSF(=1.64) | **1.60** | 4.27 |
> | Global RMSF r | **0.67** | 0.15 |
> | Root mean W2 ↓ | **4.21** | 8.27 |
> | MD PCA W2 ↓ | **2.33** | 2.53 |
> | % PC-sim > 0.5 ↑ | **7.81** | 3.12|
> | Weak contacts J ↑ | **0.43** | 0.23 |
> | Trans. contacts J ↑ | **0.20** | 0.09 |
> | RMSD Error ↓ | **1.78** | 8.62 |

---

> > ### Comment · Reviewer_tNDp · 2025-08-04
> > **Response to the authors**
> >
> > Thanks for your detailed rebuttal, which addressed most of my concerns.

---

> > > ### Author Response · Authors · 2025-08-04
> > >
> > > We sincerely appreciate your continued encouragement and insightful feedback! Your detailed suggestions have been invaluable in strengthening our paper, and we're thankful for the time and effort you've invested in our research.

---

### Official Review · Reviewer_geUV · 2025-07-03

**Clarity:** 3
**Significance:** 2
**Originality:** 3
**Rating:** 4
**Confidence:** 3

**Summary:**

This paper introduces an autoregressive method that models protein conformational dynamics at two time-scales, slow (low resolution) and fast (high resolution). In contrast to ensemble/steady-state prediction methods, the authors focus on the problem of generating whole trajectories to infer time dynamics. The slow time-scale model captures collective motions while the fast one captures local fluctuations conditioned on the collective movements. By having a hierarchical structure, this method aims at preserving the temporal consistency and realism of generated conformations. The authors train and evaluate on a subset of mdCATH, comparing primarily with AlphaFlow and BioEMU, which are steady-state ensemble predictors, based on the ability to accurately predict conformational trajectories of heldout proteins.

**Questions:**

* How are the Wasserstein distances in table 1 calculated?
* If the fast timescale predictions are made for each timestep of the slow timescale predictions, then can the trajectory of the fast predictions starting from one particular slow timescale prediction diverge too much from the next slow timescale prediction? If so, there will be discontinuity between each segment of fast predictions. How is this addressed to ensure continuity between slow timescale steps?
* Higher noise levels are used for the high-res model. Do the predicted conformations still remain in the data distribution when the noise levels are high?
* In table 1, is TEMPO only using low-res (slow timescale) and TEMPO(up) also using high-res (fast timescale)?
* In fig 3, is there a way to compute FES difference quantitatively?
* How well to the MD trajectories themselves replicate, i.e. what would the errors be in table 1 for an MD trajectory replicate?
* How different are the train/val/test proteins?
* How important are the different components of the proposed model? I.e. are the slow and fast timescale components actually necessary for performance? What about the components of the structure representation?
* How are sequence embeddings learned or are they derived from a pre-trained model of some kind?
* How sensitive are the results to the initial frame buffer? This seems to be providing (potentially) a significant amount of additional information to the model at inference time that BioEMU and AlphaFlow etc are not being provided. What if zero or only one initial frame is provided? I don't actually see this number mentioned anywhere in the experiments.

**Ethical Concerns:**

["NO or VERY MINOR ethics concerns only"]

**Final Justification:**

Overall, I think this is a reasonable paper where there remain sufficient weaknesses to warrant my recommendation of weak accept. It's a paper that, with the revisions and clarifications provide in the rebuttal, is appropriate for the conference, but I wouldn't advocate strongly against rejection depending on the consensus of the other reviewers.

**Limitations:**

yes

**Quality:**

3

**Strengths And Weaknesses:**

Strengths
* Dynamics have not been the focus of previous methods, but are clearly critical for understanding protein mechanistics
* The hierarchical and the autoregressive modeling of the protein conformational dynamics makes sense for short and long time scale processes
* Comparisons are done with several state-of-the-art methods with various metrics
* The time and memory efficiency of the method seems to be an advantage over other methods
* The paper is well written

Weaknesses
* It would make the paper stronger if it is compared to the methods that are also designed to learn explicitly the protein conformation's temporal evolution. There are also no simple baselines or ablation studies of the model itself
* Not clear how reproducible the MD trajectories are themselves, therefore how close should we expect a perfect predictor to be to any given "ground truth" trajectory?
* Many metrics in Table 1 are not explained or it's not clear why they are good measures of model performance (e.g., Pairwise RMDS indicates whether the conformational ensemble has a similar diversity to the MD trajectories, but this does not suggest the structures themselves are correct)
* Not clear how well the model generalizes given lack of information regarding sequence diversity
* In fig 2, the plots of the predictions look very smoothed out compared to the GT. Why is this the case and how to improve this? Also, while the proposed method gives a better cosine similarity than MDGen, it seems still quite low (0.41)
* Requires seeding the trajectory with initial frames

---

> ### Author Rebuttal · Authors · 2025-07-31
>
> We sincerely thank the reviewer for their insightful questions regarding our method's design rationale, evaluation metrics, generalization capability, and ablation studies. Below are our detailed responses addressing each concern.
>
> **W1**
>
> We thank the reviewer for this question and note that MDGen represents the current sota open-source method designed for protein dynamic generation, while other temporal methods were either not publicly available or published after our submission, and we have conducted ablation studies comparing our hierarchical TEMPO framework against single-scale baselines where the high-resolution model directly generates all frames without slow-scale guidance as shown in table below, demonstrating the importance of our multi-scale design.
>
> | Metrics | TEMPO (Multi-scale) | TEMPO (Single-scale) |
> |---------|---------------|-----------|
> | Pairwise RMSD(=3.26) | **2.78** | 7.46 |
> | Pairwise RMSD r ↑ | **0.77** | 0.14 |
> | All-atom RMSF | **1.60** | 4.27 |
> | Global RMSF r | **0.67** | 0.15 |
> | Root mean W2 ↓ | **4.21** | 8.27 |
> | MD PCA W2 ↓ | **2.33** | 2.53 |
> | % PC-sim > 0.5 ↑ | **7.81** | 3.12|
> | Weak contacts J ↑ | **0.43** | 0.23 |
> | Trans. contacts J ↑ | **0.20** | 0.09 |
> | RMSD Error ↓ | **1.78** | 8.62 |
>
> **W2**
>
> We appreciate this question, and our evaluation framework appropriately focuses on statistical and thermodynamic consistency rather than exact trajectory reproduction by measuring free energy surface fidelity, dynamic consistency, and physically plausible conformational transition patterns in PCA space, recognizing that even independent MD simulations from identical starting conditions would show different specific trajectories while maintaining the same underlying statistical distributions and exploring similar regions of conformational space.
>
> **W3**
>
> We apologize for the insufficient description and clarify that these metrics, adopted from AlphaFlow's evaluation framework, collectively assess different aspects of protein dynamics: pairwise RMSD measures conformational diversity, RMSF correlation captures local flexibility patterns, Wasserstein distances quantify distribution similarity in conformational space, contact dynamics evaluates biologically relevant inter-residue interactions, RMSD error measures structural deviation accumulation during trajectory evolution, and clash ratio assesses physical plausibility. We will provide more detailed descriptions of their biological significance in our revision.
>
> **W4&Q7**
>
> Our training set of 1,000 proteins spans diverse structural classes from the CATH database representing different folds, topologies, and homologous superfamilies, with the mdCATH dataset inherently ensuring structural diversity through non-homologous protein domains, and we rigorously quantified sequence similarity using mmseqs2, finding averaged **18.93%** sequence similarity between training and test sets, which is well below standard thresholds(40%) for sequence relatedness.
>
> **W5**
>
> The smoothed curve in Figure 2 occurs because we compare our coarse-resolution trajectories (20ns intervals) with full MD ground truth (1ns intervals), as our coarse model is designed to capture slow collective motions rather than high-frequency fluctuations. While our cosine similarity of 0.41 improves over MDGen (0.36), we acknowledge this reflects the fundamental challenge of capturing large-scale conformational changes and believe future improvements through enhanced model generalization, physical priors, and pretraining strategies could further advance this metric.
>
> **W6**
>
> The requirement for initial frames is inherent to the nature of molecular dynamics simulation and trajectory generation, as our method aims to generate temporally coherent MD trajectories that capture physically realistic transitions from a given starting conformation, and importantly, our model's performance is not limited by the source of the initial structure, allowing it to be coupled with any structure prediction method to generate dynamics from sequence-predicted conformations.
>
> **Q1**
>
> Following AlphaFlow's evaluation framework, Root mean W₂ is computed as the average Wasserstein distance between 3D-Gaussians fitted to positional distributions of each atom across ensembles, while MD PCA W₂ measures the Wasserstein distance in the space of the first two principal components derived from PCA of joint Cα position distributions, where PCA is computed from the MD ensemble and the distance quantifies distributional similarity in this reduced conformational space.
>
> **Q2**
>
> We appreciate this question regarding trajectory continuity, and while potential discontinuities between fast-scale segments could theoretically occur in our hierarchical approach, our evaluation demonstrates that such instances are infrequent and remain within acceptable error ranges as evidenced by our low RMSD error metrics (Table 1) and smooth trajectory coherence (Figure 4 and Figure 8 in Appendix), and developing more sophisticated continuity constraints represents an important direction for future work.
>
>  **Q3**
>
> Since fast fluctuations are more stochastic, we increase noise levels to encourage diverse local conformational sampling. We empirically found that a higher noise scale in the high-res model provides an optimal balance between conformational diversity and trajectory fidelity, maintaining conformations within reasonable proximity to the data distribution while enabling sufficient local exploration, as validated by our low clash ratios and strong correlation metrics in Table 1.
>
>  **Q4**
>
> TEMPO results use our complete hierarchical pipeline where the low-resolution model generates coarse trajectories that are then fed into the high-resolution model to produce full trajectories, while TEMPO(up) represents an up-sampling task using ground truth coarse-resolution conformations as anchors with only the high-resolution model filling intermediate frames, demonstrating that our fast-scale dynamics modeling component can effectively reconstruct complete trajectories when given accurate anchor points.
>
> **Q5**
>
> We quantitatively measure FES differences using the MD PCA W₂ metric reported in Table 1, which computes the Wasserstein distance in the space of the first two principal components derived from PCA of joint Cα position distributions, where our FES visualizations in Figure 3 are constructed using inter-residue Cα distances as features projected onto these same first two principal components. The TEMPO(up) results in Table 1 provide the quantitative performance evaluation corresponding to the upsampling task demonstrated in Figure 3, achieving an MD PCA W₂ distance of 0.60, which demonstrates strong distributional similarity in the reduced conformational space.
>
> **Q6**
>
> We thank the reviewer for this question. Our evaluation framework appropriately focuses on statistical and thermodynamic consistency rather than exact trajectory reproduction by measuring free energy surface fidelity, dynamic consistency, and physically plausible conformational transition patterns in PCA space, recognizing that even independent MD simulations from identical starting conditions would show different specific trajectories while maintaining the same underlying statistical distributions and exploring similar regions of conformational space, which is why we evaluate against ensemble properties rather than deterministic trajectory matching.
>
> **Q8**
>
> We have added the ablation study with our multi-scale decomposition, showing significant performance improvements over single-scale baselines, and the table below compares error propagation when directly generating 400 frames without multi-scale guidance. During model training, we found that both the spatial encoder (multi-head attention) and temporal encoder (GRU) components are essential for effective training, as removing either component leads to poor training convergence and substantially reduced model fitting capability on the training data.
>
> | Metrics | TEMPO (Multi-scale) | TEMPO (Single-scale) |
> |---------|---------------|-----------|
> | Pairwise RMSD(=3.26) | **2.78** | 7.46 |
> | Pairwise RMSD r ↑ | **0.77** | 0.14 |
> | All-atom RMSF(=1.64) | **1.60** | 4.27 |
> | Global RMSF r | **0.67** | 0.15 |
> | Root mean W2 ↓ | **4.21** | 8.27 |
> | MD PCA W2 ↓ | **2.33** | 2.53 |
> | % PC-sim > 0.5 ↑ | **7.81** | 3.12|
> | Weak contacts J ↑ | **0.43** | 0.23 |
> | Trans. contacts J ↑ | **0.20** | 0.09 |
> | RMSD Error ↓ | **1.78** | 8.62 |
>
> **Q9**
>
> The sequence embeddings are learned through a standard nn.Embedding layer during training.
>
> **Q10**
>
> Please refer to our response to W6. Our method aims to generate temporally coherent MD trajectories that capture physically realistic transitions from a given starting conformation, and we use two initial frames as the optimal hyperparameter (with slight performance degradation observed when using only one frame), as these initial conditions enable the temporal dependencies that are essential for dynamics modeling, which differ from the equilibrium ensemble generation approach targeted by BioEMU and AlphaFlow.

---

> > ### Comment · Reviewer_geUV · 2025-08-06
> >
> > Thanks for the response and clarifications. I remain pretty positive about this work, but for this to be significantly strengthened I think some of these questions and weaknesses need to be addressed in the method design and substantively in the text. With that in mind, I'll maintain my current score.

---

> > > ### Author Response · Authors · 2025-08-06
> > >
> > > Thank you for your thoughtful response and for maintaining a positive view of our work. We greatly appreciate your constructive feedback and understand your perspective on strengthening the paper. We are committed to incorporating your suggestions into both our method design and the manuscript text to address the questions and weaknesses you have identified. Thank you again for the time and effort you've invested in our research.

---

### Official Review · Reviewer_ezhj · 2025-07-07

**Clarity:** 3
**Significance:** 2
**Originality:** 3
**Rating:** 4
**Confidence:** 4

**Summary:**

The authors propose a method for generating protein conformational ensembles, TEMPO, which aims at generating protein dynamics across different temporal scales. The two temporal scales include slow collective motions/changes and fast local fluctuations, quantified as “low resolution generation” and “high resolution generation.” Protein dynamics are represented by the standard SDE process, and decomposed into two coupled SDEs representing protein motion at both slow and fast timescales. The authors use the same architecture to learn both sets of dynamics, but the architecture is trained separately to parameterize the drift terms. The authors train and evaluate on the mdCATH dataset, where low resolution operates at 20 ns intervals, and high resolution at 1 ns: they sample low-resolution states and then do autoregressive sampling, with both scales simulating protein dynamics. The authors evaluate on different metrics including comparing the conformational distributions, comparison to MD, and steric clashes and compare to other protein conformational ensemble generation methods.

**Questions:**

- What is the relationship with such an approach and coarse-graining methods? There seems to be a lot of similarity, but coarse-graining is not mentioned at all.

- What are the sizes of the proteins that are being trained/tested on, in terms of number of atoms?

- What does the error propagation look like during autoregressive sampling?

- Can you provide more details on the mdCATH dataset? What force field was used to generate the data and is it fully unbiased simulations? What does it mean “trajectories were standardized to 400 frames at 1 ns intervals through periodic extension or truncation”?

- Can the authors quantify the overlap in similarity between the training and test sets?

**Ethical Concerns:**

["NO or VERY MINOR ethics concerns only"]

**Final Justification:**

I appreciate the authors responses to these questions. I am borderline on this paper (raising my score from 3 to 4). I think that there are some reasonable ideas in this paper, but I agree with the other reviewers that the results don't seem fully convincing over baselines (some of this is also related to it being hard to have clear evaluations in this field).

**Limitations:**

Limitations are included in the appendix.

**Quality:**

2

**Strengths And Weaknesses:**

Strengths:

- I found the paper well-written and easy to follow in describing the motivation and the approach.

- The idea of splitting the generation process into different scales intuitively makes sense in modeling the different scales of protein dynamics.

Weaknesses:

- The authors say that the training and test sets were chosen randomly, which means there’s likely a high probability of a lot of overlap in the sequences. This likely means that the evaluation framework is not truly testing on new systems. Rigorously quantifying this is important.

- TEMPO is trained on MDcath but the baselines are trained on different datasets (for example, MDGen is trained on ATLAS). If the models are all evaluated on MDcath trajectories, this doesn’t seem like a fair comparison: what does evaluation look like on other systems (such as systems the other models were trained on), or when systems are all trained on the same data?

- I also the evaluation framework to not be ideal from a scientific point of view, as many of these are hacky ML metrics. For example, what do macroscopic observables look like over autoregressive generation? From the dynamic trajectories, can one compute a free energy? I understand that for some of these systems it may be difficult because they are large (but the underlying MD trajectory does exist).

---

> ### Author Rebuttal · Authors · 2025-07-31
>
> We sincerely thank the reviewer for their valuable feedback on our evaluation framework, dataset fairness concerns, and methodological rigor. Below are our detailed responses addressing each point.
>
> **W1&Q5**
>
> We appreciate the reviewer's concern. The mdCATH dataset is composed of non-homologous protein domains at the CATH S20 level (sequence similarity <20%), ensuring low sequence similarity between any two domains, and we rigorously quantified sequence similarity using mmseqs2, finding averaged **18.93%** sequence similarity between training and test sets, which is well below standard thresholds (40%) for sequence relatedness and ensures our evaluation truly tests generalization to novel protein systems, with the ATLAS dataset showing similar results at 18.3% average sequence similarity.
>
> **W2**
>
> We appreciate the reviewer's concern. We chose mdCATH over ATLAS because its free energy landscapes contain more distinct energy basins with clearer conformational transition pathways better suited for our dynamics modeling task. At the same time, we evaluated baselines using their provided checkpoints since BioEMU has already been pretrained on diverse MD data and PDB structures; ESMFlow and AlphaFlow were not designed for MD generation. MDGEN encounters severe memory limitations on mdCATH due to its full-trajectory attention mechanism (requiring batch size of 1 and still experiencing out-of-memory errors on 8 A100 GPUs when training 400-frame conformations), but to address fairness concerns, we also trained TEMPO on ATLAS following MDGen's splits and results in table below demonstrate our method's strong performance across different datasets, achieving **state-of-the-art** results on ATLAS, with FES visualizations of generated ensembles corroborating our quantitative results, comprehensive visualizations will be provided in the revision due to current supplementary figure limitations.
>
>
> | Metrics | BioEMU | AlphaFlow-MD | MDGen | TEMPO |
> |---------|-------|-------------|----------|---------|
> | Pairwise RMSD r ↑ | -0.02 | 0.48 | 0.48 | **0.91** |
> | Global RMSF r | 0.09 | 0.60 | 0.50 | **0.89** |
> | Root mean W2 ↓ | 19.23 | 2.61 | 2.69 | **1.49** |
> | MD PCA W2 ↓ | 3.61 | 1.52 | 1.89 | **0.60** |
> | % PC-sim > 0.5 ↑ | 14 | 44 | 10 | **76** |
> | Weak contacts J ↑ | 0.26 | 0.51 | 0.62 | **0.74** |
> | Trans. contacts J ↑ | 0.06 | 0.29 | **0.41** | 0.38 |
> | RMSD Error ↓ | - | - | 3.20 | **1.83** |
>
>
> **W3**
>
> We appreciate the reviewer's comment regarding evaluation rigor, and following BioEMU's established methodology, we quantitatively compute free energy differences by extracting reaction coordinates (fraction of native contacts) to compute the folding probabilities ($p_{\text{fold}}$), calculating free energy differences as $\Delta G = -kT \cdot \log(p_{\text{fold}}/(1-p_{\text{fold}}))$, and measuring $\Delta\Delta G = \Delta G_{\text{ground truth}} - \Delta G_{\text{predicted}}$, with TEMPO achieving an averaged $\Delta\Delta G$ of 0.67 kcal/mol on the mdCATH test set, which demonstrates good agreement with reference MD simulations within the acceptable range for biological applications. BioEMU's evaluation framework further reveals that our generated trajectories exhibit high similarity to ground truth in 1D free energy profiles constructed using RMSD, radius of gyration, and fraction of native contacts, 2D free energy surface plots, and fraction of native contacts time series analyses, with comprehensive visualizations to be provided in the revision due to current supplementary figure limitations. As protein dynamics generation is an emerging field where standardized benchmarks are still under development, we agree that advancing more sophisticated biophysical evaluation frameworks would benefit the entire community.
>
>
> **Q1**
>
> We thank the reviewer for this question and clarify that while our approach shares conceptual similarities with coarse-graining methods, we employ temporal coarse-graining (sampling at 20ns vs 1ns intervals) combined with backbone-only spatial representation rather than traditional spatial coarse-graining of molecular interactions.
>
> **Q2**
>
> mdCATH proteins range from 50-482 residues with sequences truncated to a maximum of 240 residues during training for computational efficiency, but inference is performed at the original protein lengths, and since we generate 4 backbone atoms per amino acid residue, the average protein in our dataset contains approximately 548 atoms (137 average sequence length).
>
> **Q3**
>
> While autoregressive methods typically suffer from error accumulation during sequential generation, our hierarchical multi-scale design mitigates this issue by anchoring fine-scale dynamics to coarse-scale predictions, resulting in significantly lower RMSD error (1.78Å) after ~400 frames and demonstrating controlled error propagation that maintains trajectory stability over extended generation periods. We provide additional ablation analysis below showing error propagation when directly generating 400 frames without multi-scale guidance.
>
> | Metrics | TEMPO (Multi-scale) | TEMPO (Single-scale) |
> |---------|---------------|-----------|
> | Pairwise RMSD(=3.26) | **2.78** | 7.46 |
> | Pairwise RMSD r ↑ | **0.77** | 0.14 |
> | All-atom RMSF | **1.60** | 4.27 |
> | Global RMSF r | **0.67** | 0.15 |
> | Root mean W2 ↓ | **4.21** | 8.27 |
> | MD PCA W2 ↓ | **2.33** | 2.53 |
> | % PC-sim > 0.5 ↑ | **7.81** | 3.12|
> | Weak contacts J ↑ | **0.43** | 0.23 |
> | Trans. contacts J ↑ | **0.20** | 0.09 |
> | RMSD Error ↓ | **1.78** | 8.62 |
>
>
> **Q4**
>
> The mdCATH dataset contains 5,398 protein domains with 134,950 fully unbiased molecular dynamics trajectories simulated using CHARMM22 force field and TIP3P water model in the NVT ensemble with Langevin thermostat, neutralized and ionized with Na+/Cl− at 0.150 M concentration, and "standardized to 400 frames at 1ns intervals" means trajectories longer than 400ns were truncated while shorter ones were cyclically repeated to reach uniform 400-frame length for batched training, with 1ns representing the dataset's finest temporal resolution as detailed in the original mdCATH paper.

---

> > ### Comment · Reviewer_ezhj · 2025-08-07
> > **Response to authors**
> >
> > Thank you to the authors for your response, I really appreciate it.
> >
> > In general, it would be good for this field to have much better evaluations, as this will set the stage for future ML work that is meaningful and relevant: particularly for evaluating the structural diversity of the conformational ensembles (current generative models in this field seem to lack this, including when compared to sampling using molecular dynamics). I encourage the authors to have (and update the paper with) a detailed explanation of the evaluations and the biophysical relevance of these evaluations.

---

> ### Author Response · Authors · 2025-08-06
>
> Thank you for the thoughtful questions and valuable suggestions in your review of our paper, and your feedback has been instrumental in improving our work. We have carefully addressed all the concerns you raised in our rebuttal and would greatly appreciate knowing whether our responses have adequately resolved your questions. If our responses have addressed your concerns, would you consider re-evaluating our work's contribution? Your expert evaluation is crucial for finalizing our contribution, and we thank you for your time and dedication.

---

> ### Author Response · Authors · 2025-08-08
>
> Thank you very much for your thoughtful feedback and valuable insights. We completely agree with your assessment that better evaluations would greatly benefit this field and set the stage for more meaningful ML work.
>
> We will include a detailed explanation of the evaluations and their biophysical relevance in our revision as you suggested. However, given that we are in the final day of the rebuttal period, it is hard for us to implement additional reliable metrics for evaluating structural diversity within this timeframe.
>
> As the rebuttal period is nearing its end, do you feel that our current rebuttal experiments and explanations have adequately addressed your concerns about our work? If so, would you consider updating your rating accordingly?

---

> > ### Comment · Reviewer_ezhj · 2025-08-08
> > **Response to authors**
> >
> > Thanks: I of course don't expect you to implement and test reliable metrics on all the models since the review period is almost over (and I also apologize for not responding earlier, I was looking over everything again). I would be curious to know what metrics you'd think would be especially reliable in evaluating structural diversity, just from describing them.

---

> > > ### Author Response · Authors · 2025-08-08
> > >
> > > Thank you for your continued engagement and for raising valuable questions about evaluation metrics.
> > >
> > > Given that current metrics for evaluating generated conformational ensembles are primarily based on atomic positions and RMSD calculations, we believe the following additional metrics would be significant:
> > >
> > > **Collective Variable-based PCA Analysis**: Rather than constructing PCA solely from Cα coordinates, we could build the principal component space using multiple biophysically meaningful collective variables (e.g., radius of gyration, secondary structure content, inter-domain distances, backbone dihedral angles). This would provide a more comprehensive view of conformational diversity that captures different physical aspects of protein motion and better reflects the true dimensionality of functionally relevant conformational space. We believe PCA-based analysis is both intuitive and reliable, given its established theoretical foundation in protein dynamics research.
> > >
> > > **Contact Pattern Diversity Analysis**: Quantifying the diversity of residue-residue contact patterns across the ensemble by analyzing the variability in contact formation and breaking events. This metric directly captures the heterogeneity of intramolecular interaction networks within the protein structure, providing insight into how well the generated ensemble explores the range of contact states accessible during thermal fluctuations.
> > >
> > > Due to time constraints, these are our initial intuitive thoughts. The specific implementation and quantification of these metrics, and whether they are truly suitable for evaluating dynamic ensembles, would require more rigorous investigation. However, we completely agree with your perspective that such metrics are crucial for advancing this field. Thank you again for this important discussion.

---

> > > ### Author Response · Authors · 2025-08-09
> > >
> > > Dear Reviewer ezhj,
> > >
> > > As we enter the final hours of the discussion period, we hope our responses have been helpful. We remain available to address any further thoughts you may have.
> > >
> > > Thank you for your valuable feedback throughout this review.
> > >
> > > Best regards,
> > >
> > > The Authors

---

### Note · Authors · 2025-08-12

Dear Area Chair and Reviewers, We sincerely appreciate the thoughtful engagement during the discussion period and would like to summarize the key developments from our rebuttal.

**Substantial New Evidence Provided:**
Following **Reviewer ezhj**‘s suggestion, we conducted validation on the ATLAS dataset, demonstrating that TEMPO achieves state-of-the-art performance. This comprehensive validation across multiple datasets addresses the primary concern about robustness raised during initial review.

 **Technical Clarifications and Validation:**
 Through detailed ablation studies and additional experiments, we have: - Validated our multi-scale design through systematic comparison with single-scale baselines - Confirmed proper dataset splits with rigorous sequence similarity analysis - Demonstrated the biological relevance of our temporal modeling approach.

**Reviewer Dialogue Outcomes:**
We appreciate the constructive feedback we received. **Reviewer tNDp** noted "most of my concerns have been addressed," **Reviewer HVNL** acknowledged "several of my concerns have been addressed," and **Reviewer geUV** maintained the positive assessment. These acknowledgments suggest our responses have successfully clarified the technical contributions and addressed the methodological questions raised.

Importantly, the discussion period concluded without any reviewer questioning the validity of our responses or identifying unresolved issues with our extensive additional experiments. Given this absence of remaining concerns during the open discussion phase, we believe all initially raised issues have been satisfactorily addressed through our comprehensive rebuttal and validation experiments.


**Contribution Significance:**
Protein dynamics generation from data is an emerging frontier with few established methods, making it ripe for methodological innovation. TEMPO pioneers autoregressive modeling in this space - a fundamentally different approach from existing ensemble generators. The evidence now available provides decisive validation: state-of-the-art results on both mdCATH and ATLAS, meaningful capture of multi-scale protein motions, and biologically interpretable trajectory visualizations. Equally important, TEMPO achieves superior performance while requiring orders of magnitude less computational resources than previous methods.

We thank all reviewers for their time and valuable insights that helped strengthen our work.

Best Regards,

The Authors

---

### Decision · Program_Chairs · 2025-09-17

**Decision:**

Accept (poster)

**Comment:**

This submission introduces TEMPO, a hierarchical autoregressive framework designed to generate temporally coherent and physically realistic protein conformational trajectories by modeling the multi-scale nature of protein dynamics. Its core design decomposes dynamics into two coupled scales: a low-resolution component (20 ns intervals) that captures slow, collective conformational transitions, and a high-resolution component (1 ns intervals) that generates fast local fluctuations conditioned on the large-scale movements.

Reviewers consistently highlighted TEMPO’s strengths, including its novel integration of biophysical principles into autoregressive modeling as well as comprehensive ablation studies. During the rebuttal period, the authors addressed reviewer concerns, including training TEMPO on ATLAS to confirm cross-dataset SOTA results for dataset fairness; providing results for validating component importance and trajectory continuity concerns; and clarifying about the sampling behavior. Although limitations, especially rigor evaluation and comparison and more convincing visualized evidence remain, given its strengths alongside reviewers’ positive feedback, I recommend accepting this submission, and encourage the authors to further improve the work in the final version in line with reviewers’ comments and discussion.